# On the Adversarial Robustness of Large Vision-Language Models under Visual Token Compression

**Xinwei Zhang** [1]  **Hangcheng Liu** [2]  **Li Bai** [1]  **Hao Wang** [3]  **Qingqing Ye** [1]  **Tianwei Zhang** [2]  **Haibo Hu** [1 4]

## Abstract

Visual token compression is widely used to accelerate large vision-language models (LVLMs) by pruning or merging visual tokens, yet its adversarial robustness remains unexplored. We show that existing encoder-based attacks cannot fully disclose the robustness vulnerabilities of compressed LVLMs, due to an optimization-inference mismatch: perturbations are optimized on the full-token representation, while inference is performed through a token-compression bottleneck. To address this gap, we propose the **C**ompression-**A**li**G**n**E**d attack (CAGE), which aligns perturbation optimization with compression inference without assuming access to the deployed compression mechanism or its token budget. CAGE combines (i) *expected feature disruption*, which concentrates distortion on tokens likely to survive across plausible budgets, and (ii) *rank distortion alignment*, which actively aligns token distortions with rank scores to promote the retention of highly distorted evidence. Across diverse representative plug-and-play compression mechanisms and datasets, our results show that CAGE consistently achieves lower robust accuracy than the baseline. This work highlights that robustness assessments ignoring compression can be overly optimistic, calling for compression-aware security evaluation and defenses for efficient LVLMs.

## 1. Introduction

The rapid advancement of large vision-language models (LVLMs) has revolutionized multimodal understanding, en-

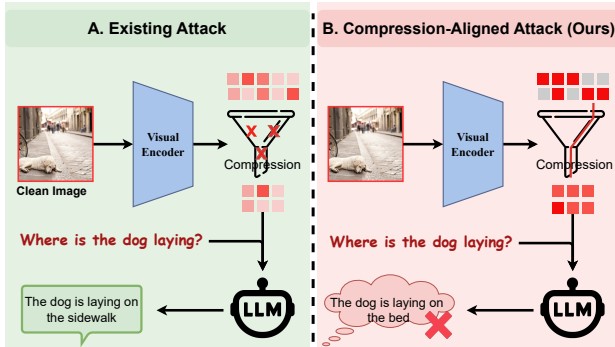

*Figure 1.* Comparison between the existing attack and our attack. Darker red indicates tokens with stronger adversarial perturbation. While the existing attack (A) perturbs all visual tokens (all tokens are red), CAGE (B) concentrates the distortion on the surviving tokens (only survivors are red).

abling remarkable capabilities in tasks ranging from visual question answering to complex reasoning (Liu et al., 2023; OpenAI et al., 2024; Team et al., 2025). However, this progress comes with a substantial computational burden: current state-of-the-art models like LLaVA-NeXT (Liu et al., 2024a) and InternVL (Chen et al., 2024b) process hundreds to thousands of visual tokens per image, creating significant efficiency bottlenecks that limit their practical deployment. This bottleneck is further amplified in agentic settings (especially mobile agents), where the model must process multiple images under tight latency and energy budgets (Zhang et al., 2025a; Ye et al., 2025).

Given these efficiency constraints, recent research has focused intensively on visual token compression (Liu et al., 2024b; Shao et al., 2025; Bolya et al., 2023; Alvar et al., 2025; Tong et al., 2025). In particular, plug-and-play methods such as VisionZip (Yang et al., 2025) and Vis-Pruner (Zhang et al., 2025b) identify a compact set of "informative" tokens using importance scores (e.g., attention-based saliency), and optionally merge additional tokens via similarity-based merging before passing the compressed visual sequence to the language model. By reducing the visual-token budget at inference time, these approaches deliver substantial efficiency gains while largely preserving accuracy on normal tasks. However, the adversarial robust-

[1]The Hong Kong Polytechnic University, Hong Kong [2]Nanyang Technological University, Singapore [3]Chongqing University, Chongqing, China [4]Research Centre for Privacy and Security Technologies in Future Smart Systems, PolyU. Correspondence to: Li Bai <baili.bai@connect.polyu.hk>.

*Proceedings of the 43rd International Conference on Machine Learning*, Seoul, South Korea. PMLR 306, 2026. Copyright 2026 by the author(s).

ness of these compressed LVLMs remains unexplored.

Assessing the adversarial robustness of such models is crucial, as token compression has become indispensable for deploying LVLMs in real-time, safety-critical applications (e.g., autonomous driving and robotics) due to strict latency and energy constraints (Lübberstedt et al., 2025; Hu et al., 2025; Guerrero et al., 2025). Existing evaluations typically consider encoder-based attacks, which optimize perturbations over global visual tokens (Mei et al., 2026; Wang et al., 2024c). This choice is pragmatic as encoder-only optimization is far cheaper than end-to-end attacks, and vision encoders are often publicly available even when the full LVLM is not (Mei et al., 2026; Cui et al., 2024; Wang et al., 2024c). However, as illustrated in Figure 1, existing attacks cannot fully disclose the robustness vulnerabilities of compressed LVLMs due to an *optimization-inference mismatch*: perturbations are optimized in the full-token space, whereas model inference operates on the compressed token representation.

To address this mismatch, we propose the **C**ompression-**A**li**GnE**d attack (`CAGE`), an adversarial framework that aligns perturbation optimization with the compressed token space without assuming knowledge of the deployed compression mechanism or configuration (e.g., token budget). The core challenge lies in the uncertainty of deployment budgets, which renders the effective attack feature space unpredictable during optimization. To overcome it, `CAGE` couples two complementary objectives: (i) *Expected Feature Disruption* (EFD), which models the unknown token budget probabilistically and concentrates distortion on tokens likely to survive the bottleneck; and (ii) *Rank Distortion Alignment* (RDA), which aligns the ranking signal (e.g., attention scores) with distortion strength, so that the most perturbed tokens also have the highest probability of being selected.

We evaluate `CAGE` across five representative compression mechanisms and three datasets. Our results show that `CAGE` consistently yields substantially lower adversarial accuracy than the existing attack. In addition, we investigate some possible defenses. While both defenses show somewhat encouraging results, they are not yet sufficient, highlighting the need for more effective defenses.

In summary, the contributions of this paper are as follows.

- We are the *first* to explore the adversarial robustness of LVLMs under visual token compression.
- We identify an optimization-inference mismatch between existing attacks and compressed inference, which leads to an overestimation of robustness.
- We propose a novel attack, named `CAGE`[1], that remains

---

[1]Source code is available at https://github.com/XinweiZhang1998/CAGE.

effective under unknown compression mechanisms and varying deployment token budgets.
- We demonstrate the effectiveness of our attack across diverse compression mechanisms, token budgets, and datasets, and further explore potential defense strategies.

## 2. Preliminaries

### 2.1. $K$-Compressed LVLM

A standard LVLM typically consists of a visual encoder $\mathcal{E}$, a visual projector $\mathcal{P}$, and a large language model (LLM) $\mathcal{F}$. Given an input image $\mathbf{x}_v \in \mathbb{R}^{H \times W \times C}$ and a textual prompt $\mathbf{x}_t$ (e.g., a question in VQA tasks), the visual encoder extracts a sequence of visual tokens $\mathbf{H} = \mathcal{E}(\mathbf{x}_v) \in \mathbb{R}^{N \times D}$, where $N$ is the number of tokens (typically $N = 576$ or higher) and $D$ is the embedding dimension.

In a $K$-compressed LVLM, a compression module $\mathcal{C}(\cdot; K)$ is introduced to reduce the visual sequence length from $N$ to a smaller budget $K$ ($K < N$) before feeding them into the LLM. In the context of model inference, we refer to this constraint $K$ as the **deployment budget** (denoted as $K_{\text{model}}$). This process can be formulated as:

$$\mathbf{Z} = \mathcal{C}(\mathbf{H}; K) \in \mathbb{R}^{K \times D}, \tag{1}$$

where $\mathcal{C}$ represents the compression strategy (e.g., token selection/pruning or merging based on attention scores). The compressed visual tokens $\mathbf{Z}$ are then concatenated with the text embeddings of $\mathbf{x}_t$ and fed into the LLM to generate the response $\mathbf{Y}$. The probability of generating the response is modeled auto-regressively:

$$P(\mathbf{Y}|\mathbf{x}_v, \mathbf{x}_t) = \prod_{i=1}^{L} P(y_i|\mathbf{Z}, \mathbf{x}_t, y_{<i}), \tag{2}$$

where $y_{<i}$ denotes the tokens generated before step $i$.

### 2.2. Threat Model

In this paper, we focus on the adversarial robustness of LVLMs under a realistic gray-box threat model, following current attacks (Wang et al., 2024c; Mei et al., 2026).

**Knowledge Assumption.** We assume that the adversary has white-box access to the visual encoder $\mathcal{E}$, as such backbones are often open-sourced in practice (Wang et al., 2024a;c; Mei et al., 2026). In contrast, the adversary has black-box access to both the compression module $\mathcal{C}$ and downstream LLM $\mathcal{F}$. Specifically, we treat compression hyperparameters (e.g., the deployment budget $K_{\text{model}}$) as unknown runtime variables. This reflects common deployments where token compression is diverse and often customized by the model owner, and the downstream LLM is typically proprietary and not publicly available.

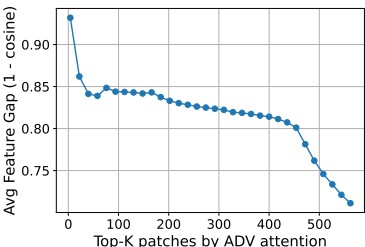

*Figure 2.* Average token-level feature gap under VEAttack over 100 samples. We rank vision tokens by adversarial(ADV) attention and plot the average feature gap $(1 - \text{cosine})$ over the top-$K$ tokens. The curve shows that the gap is large on a small number of high-attention tokens and gradually decreases as lower-ranked tokens are included.

**Attack Objective.** Given an image $\mathbf{x}_v$, question $\mathbf{x}_t$, and ground-truth $\mathbf{Y}_{gt}$, the adversary seeks an imperceptible perturbation $\boldsymbol{\delta}$ to mislead the prediction. Ideally, the goal is to maximize the negative log-likelihood of the ground-truth answer:

$$\mathcal{L}(\boldsymbol{\delta}) = -\log P(\mathbf{Y}_{gt} | \mathcal{C}(\mathcal{E}(\mathbf{x}_v + \boldsymbol{\delta}); K), \mathbf{x}_t). \quad (3)$$

Under our gray-box setting, backpropagating through the full surrogate LVLM is computationally expensive and prone to overfitting. Therefore, following prior work (e.g., VEAttack (Mei et al., 2026)), we adopt an encoder-based attack formulation. Instead of optimizing the output-text probability, the adversary directly perturbs the vision encoder's representations under white-box access, making the attack task- and question-agnostic. Specifically, we maximize the semantic deviation by maximizing the cosine distance between clean features $\mathbf{H} = \mathcal{E}(\mathbf{x}_v)$ and adversarial features $\mathbf{H}' = \mathcal{E}(\mathbf{x}_v + \boldsymbol{\delta})$:

$$\max_{\boldsymbol{\delta}} \left(1 - \mathcal{S}(\mathbf{H}, \mathbf{H}')\right), \quad (4)$$

where $\mathcal{S}(\mathbf{u}, \mathbf{v}) = \frac{\mathbf{u}^\top \mathbf{v}}{\|\mathbf{u}\|\|\mathbf{v}\|}$ denotes the cosine similarity between two vectors. Since we perturb only the visual input, we will denote $\mathbf{x}_v$ simply as $\mathbf{x}$ in the following for notational simplicity. By pushing adversarial representations away from the clean feature in the visual embedding space, the attack induces erroneous downstream responses without requiring access to the specific prompt.

## 3. Motivation: An Empirical Study

To understand how visual token compression impacts the adversarial robustness of LVLMs, we conduct an empirical study by applying VEAttack (Mei et al., 2026) to LLaVA equipped with VisionZip (Yang et al., 2025). VisionZip reduces the visual token budget via Top-$K$ token selection followed by token aggregation, a common design pattern in recent compressed LVLMs. Our analysis yields two key insights as follows.

*Table 1.* Robust accuracy (%, ↓) on compressed LLaVA when varying the model's token budget $K_{\text{model}}$ (rows) and the attack optimization budget $K_{\text{attack}}$ (columns). For each deployment setting, the most harmful configuration (in **bold**) typically corresponds to an attack budget aligned with the model's effective token budget, rather than the full-token baseline.

| $K_{\text{attack}}$ / $K_{\text{model}}$ | 576 (Full) | 192 | 64 | 16 |
|---|---|---|---|---|
| **192** | 55.7 | **50.7** | 53.2 | 60.4 |
| **64** | 53.5 | 48.8 | **48.7** | 56.0 |
| **16** | 49.7 | 45.3 | 44.7 | **44.4** |

We first investigate whether token pruning inadvertently mitigates adversarial effects by filtering out perturbed tokens. Figure 2 probes the token-level representation drift by ranking visual tokens according to adversarial attention scores and computing the average feature gap (cosine distance) between clean and adversarial features over the top-$K$ tokens. We observe that the average feature gap is larger for smaller $K$ and monotonically decreases as $K$ increases. This implies that adversarial distortion is concentrated on a small set of high-importance tokens and is diluted as lower-ranked unimportant tokens are included. Since the compression mechanism preserves high-importance tokens, it keeps the most distorted tokens while discarding weakly perturbed low-importance ones, yielding a "distortion-concentrated" survivor set. Consequently, the compressed model bases its prediction on the most strongly corrupted visual evidence rather than a denoised representation.

> **Insight I**: *Under adversarial inputs, compression behaves as a "distortion concentrator" rather than a denoiser: it preferentially retains heavily corrupted, high-importance tokens while discarding relatively clean, low-importance tokens.*

While Insight I shows that compression selects heavily distorted survivor tokens, we next investigate whether existing attacks remain maximally harmful under such compressed inference. To probe this, we introduce an attack optimization budget $K_{\text{attack}}$ and modify the VEAttack objective to compute the loss only on the Top-$K_{\text{attack}}$ visual tokens. Let $\text{TOP}_{K_{\text{attack}}}(\cdot)$ denote the subset of $K_{\text{attack}}$ token features with the largest attention scores, where the attention scores are computed on the current adversarial forward pass and updated at each optimization step. We optimize

$$\max_{\boldsymbol{\delta}} \left(1 - \mathcal{S}\left(\text{TOP}_{K_{\text{attack}}}(\mathbf{H}), \text{TOP}_{K_{\text{attack}}}(\mathbf{H}')\right)\right). \quad (5)$$

As shown in Table 1, the standard full-token setting ($K_{\text{attack}}$=Full) tends to be overly optimistic under compressed inference: restricting optimization to a compression-aware token budget yields lower robust accuracy. This indicates a clear **optimization-inference mismatch** in existing attacks: perturbations are optimized over the full token

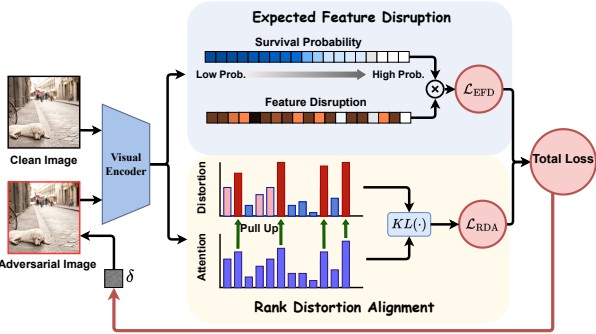

*Figure 3.* Overview of CAGE.

space, whereas inference depends on a compressed token representation. Under compression, existing attacks that optimized on full token space can become suboptimal due to: (i) *Budget Dilution*: a non-trivial portion of the optimization signal is allocated to tokens that are later pruned and therefore never influence inference; and (ii) *Dependency Disruption*: pruning context/background tokens can alter cross-token interactions, potentially weakening the attack effect that global objectives rely on.

Moreover, attack effectiveness depends on budget alignment: attacks are typically strongest when $K_{attack}$ is comparable to the deployment budget $K_{model}$. For example, on the 16-token model, the full-token attack yields 49.7% robust accuracy, whereas the aligned setting ($K_{attack}$=16) reduces it to 44.4%. This motivates us to align attack optimization with the post-compression feature space that the deployed model actually uses.

> *Insight II: Token compression creates an optimization-inference mismatch, making existing attacks suboptimal and overly optimistic.*

**In summary**, our analysis highlights a critical gap in robustness evaluation for token-compressed LVLMs. Compression forces the model to rely on heavily distorted survivor tokens (Insight I), yet existing attacks optimize for the full-token space, ignoring the post-compression inference pathway. This optimization-inference mismatch (Insight II) results in optimistic robustness estimates. These findings motivate us to propose an attack that aligns perturbation optimization with the token compression mechanism.

## 4. Compression-Aligned Attack

Motivated by the findings in Section 3, we propose a compression-aligned attack (named CAGE). CAGE bridges the optimization-inference mismatch by reformulating the adversarial objective: instead of indiscriminately disrupting global visual features, CAGE targets the compressed representation that the victim model actually consumes at

inference time. Concretely, CAGE concentrates the perturbation on the limited set of survivor tokens that pass through the token-selection bottleneck, without assuming knowledge of the deployment token budget.

### 4.1. Challenge and Overview

**Key challenge.** An effective attack must concentrate perturbations on the survivor tokens that pass the token-selection bottleneck. However, the deployment token budget $K_{model}$ is unknown and varies across models. This uncertainty makes the attack budget $K_{attack}$ non-trivial: underestimating $K_{model}$ leaves part of the survivor set unperturbed (incomplete coverage), whereas over-estimating $K_{model}$ allocates optimization signal to tokens that will not survive selection (budget dilution), as presented in Table 1.

**Overview.** To address the above challenge, we propose CAGE, a probabilistic attack framework designed to optimize perturbations through an uncertain compression bottleneck. In CAGE, we strategically target the token selection stage because it is a shared first step in general compression pipelines: some methods perform selection only (Arif et al., 2025; Alvar et al., 2025), while others first select representative tokens and then merge remaining tokens into them (Yang et al., 2025; Shang et al., 2025). Attacking this selection stage provides a unified interface that naturally covers currently used compression mechanisms. As shown in Figure 3, CAGE mainly includes two components:

- *Expected Feature Disruption (EFD)* maximizes the expected feature disruption that is propagated through compression. It models the deployment budget $K_{model}$ as a random variable and reweights the distortion of each token by its retention probability, focusing on the tokens likely to survive the bottleneck.
- *Rank Distortion Alignment (RDA)* aligns selection scores (e.g., attention) with token distortions via a differentiable distribution-matching loss. This increases the probability that highly distorted tokens are ranked within the selected set and thus influence the compressed representation.

### 4.2. Expected Feature Disruption

A naive way to align the attack with the compression bottleneck is to weight token-level feature disruption by the importance scores $s_i(\mathbf{x})$. However, this is often suboptimal because attention scores are typically highly peaky: most mass concentrates on a few top-ranked tokens even after normalization. As a result, $s_i(\mathbf{x})$-weighted objectives tend to over-focus on the very top ranks and provide little optimization signal for intermediate tokens. This is undesirable because a non-trivial portion of these intermediate tokens can still be selected under compression, especially when the deployment budget is moderate or unknown.

To address this, we instead focus on each token's survival probability. Since the deployment budget $K_{\text{model}}$ is unknown, we model it as a discrete random variable drawn from a prior distribution $P(K_{\text{model}})$ (e.g., $K_{\text{model}} \sim \mathcal{U}[K_{\text{min}}, K_{\text{max}}]$). Let $r_i(\mathbf{x})$ denote the rank of token $i$ obtained by sorting selection scores $s_i(\mathbf{x})$ in descending order (with rank starting from 0). Under Top-$K$ selection, token $i$ is retained if and only if $r_i(\mathbf{x}) < K_{\text{model}}$. We thus define its *survival probability* as:

$$
\begin{aligned}
\pi_i(\mathbf{x}) &= P(K_{\text{model}} > r_i(\mathbf{x})) \\
&= \sum_k P(K_{\text{model}}=k) \cdot \mathbb{I}(r_i(\mathbf{x}) < k),
\end{aligned} \quad (6)
$$

where $\mathbb{I}(\cdot)$ denotes the indicator function. Under a uniform prior, $\pi_i(\mathbf{x})$ forms a rank-decaying soft mask: it equals 1 for top-ranked tokens ($r_i < K_{\text{min}}$), gradually decays for intermediate ranks, and becomes 0 for low-ranked tokens ($r_i \geq K_{\text{max}}$). This allows us to emphasize tokens that are consistently retained across plausible deployment budgets, without committing to a single fixed $K_{\text{model}}$ during attack optimization.

We then use $\pi_i(\mathbf{x})$ to define an objective that maximizes the feature disruption propagated through the compression bottleneck. Let $\mathbf{z}_i^{\text{cln}}$ and $\mathbf{z}_i^{\text{adv}}$ denote the clean and adversarial feature vectors of token $i$ at the selection layer. We quantify token-level disruption using cosine distance:

$$
d_i(\mathbf{x}) = 1 - \mathcal{S}\big(\mathbf{z}_i^{\text{adv}}, \mathbf{z}_i^{\text{cln}}\big). \quad (7)
$$

The EFD loss is computed as the probability-weighted average distortion:

$$
\mathcal{L}_{\text{EFD}}(\mathbf{x}) = \frac{\sum_{i=1}^{N} \pi_i(\mathbf{x}) \cdot d_i(\mathbf{x})}{\sum_{i=1}^{N} \pi_i(\mathbf{x})}. \quad (8)
$$

By weighting distortions with $\pi_i(\mathbf{x})$, $\mathcal{L}_{\text{EFD}}$ concentrates perturbation energy exclusively on high-probability survivors. This formulation simultaneously addresses the twin failure modes identified in Section 3: It mitigates *budget dilution* by ignoring tokens destined for pruning (where $\pi_i \to 0$), and circumvents *dependency disruption* by optimizing the adversarial representation directly within the compressed view, preventing the attack from relying on fragile interactions with vanishing background context.

### 4.3. Rank Distortion Alignment

While EFD concentrates perturbations on tokens that are likely to survive compression, it does not explicitly align selection with perturbation strength. Ideally, we want a state of rank-distortion alignment, where the tokens carrying the strongest adversarial perturbations are also assigned the highest selection scores to guarantee their retention.

**Why EFD is not enough?** In theory, EFD would implicitly encourage such alignment. Applying the chain rule to Equation (8) yields

$$
\nabla_{\mathbf{x}} \mathcal{L}_{\text{EFD}} \propto \sum_{i=1}^{N} \underbrace{\pi_i \cdot \nabla_{\mathbf{x}} d_i}_{\text{survivor corruption}} + \underbrace{d_i \cdot \nabla_{\mathbf{x}} \pi_i}_{\text{distortion promotion}}. \quad (9)
$$

The first term, $\sum_i \pi_i \nabla_{\mathbf{x}} d_i$, increases distortion on tokens weighted by their current retention probabilities, i.e., it corrupts the tokens that are already likely to be retained. The second term, $\sum_i d_i \nabla_{\mathbf{x}} \pi_i$, would encourage rank-distortion alignment by pushing the retention weights toward highly distorted tokens, making them more likely to be selected. However, $\pi_i(\mathbf{x})$ is induced by discrete ranking and Top-$K$ selection, which introduce non-smooth, piecewise-constant dependencies on the underlying scores. Backpropagating through $\pi_i$ typically yields sparse and unstable gradients (and can be ill-defined at switching points), providing only a limited signal for consistently promoting highly distorted tokens in the ranking. Consequently, optimization mainly increases distortion on tokens that are already selected/likely retained, but does not reliably steer the selection mechanism toward the most distorted tokens.

To explicitly enforce alignment, we introduce a differentiable *Rank Distortion Alignment (RDA)* objective. We convert $d_i$ and $s_i^{\text{adv}}$ into token-wise probability distributions via a softmax:

$$
p_i^{(d)}(\mathbf{x}) = \frac{\exp(d_i(\mathbf{x}))}{\sum_j \exp(d_j(\mathbf{x}))}, \; p_i^{(s)}(\mathbf{x}) = \frac{\exp(s_i^{\text{adv}}(\mathbf{x}))}{\sum_j \exp(s_j^{\text{adv}}(\mathbf{x}))}. \quad (10)
$$

We then align the selection mechanism to the distortion profile by maximizing the expected log-likelihood of the selection distribution with respect to the distortion target:

$$
\mathcal{L}_{\text{RDA}}(\mathbf{x}) = \sum_{i=1}^{N} p_i^{(d)}(\mathbf{x}) \log p_i^{(s)}(\mathbf{x}). \quad (11)
$$

During optimization, we treat $p^{(d)}$ as a fixed target and stop gradients through $p^{(d)}$ to avoid degenerate dynamics where both distributions drift simultaneously. This encourages the selection mechanism to prioritize highly distorted tokens, increasing the chance that adversarial evidence is propagated through the compression bottleneck.

### 4.4. Joint Optimization

We generate adversarial perturbations $\boldsymbol{\delta}$ using projected gradient descent (PGD) under an $\ell_\infty$ constraint. At each iteration, we compute: (i) the survival probabilities $\pi_i$ from the current adversarial selection scores, (ii) the expected feature disruption $\mathcal{L}_{\text{EFD}}$ via Equation (8), and (iii) the rank-distortion alignment objective $\mathcal{L}_{\text{RDA}}$ via Equation (11). The

total optimized objective is given by:

$$\max_{\boldsymbol{\delta}} \quad \mathcal{L}_{\text{total}} = \mathcal{L}_{\text{EFD}} + \lambda \cdot \mathcal{L}_{\text{RDA}}, \qquad \text{s.t. } \|\boldsymbol{\delta}\|_{\infty} \le \epsilon, \tag{12}$$

where $\lambda$ is a hyperparameter that balances $\mathcal{L}_{\text{EFD}}$ and $\mathcal{L}_{\text{RDA}}$. Intuitively, $\mathcal{L}_{\text{EFD}}$ produces high-distortion features on likely survivors (payload), while $\mathcal{L}_{\text{RDA}}$ ensures that the model prioritizes these distorted tokens during selection (delivery), yielding a robust attack on the compressed representation across varying and unknown budgets. The whole procedure is given in Algorithm 1 in Appendix.

## 5. Experiments

### 5.1. Experimental Setup

**Victim Models.** Our evaluation primarily focuses on the widely adopted LLaVA (Liu et al., 2023), given its prevalent use as a backbone model in existing token compression research. To evaluate the cross-model generalization of our attack, we also conduct experiments on Qwen2.5-VL (Bai et al., 2025c) with the results in Appendix D.

**Compression Mechanisms.** To ensure broad coverage of compression mechanisms, we evaluate five diverse methods: VisionZIP (Yang et al., 2025), VisPruner (Zhang et al., 2025b), DivPrune (Alvar et al., 2025), FlowCut (Tong et al., 2025), and PruMerge (Shang et al., 2025). The detailed descriptions of these mechanisms are given in Appendix C.2. We assess these methods across a broad spectrum of deployment token budgets: (i) *Full Sequence (Upper Bound):* no compression, retaining all $K_{\text{model}}$=576 tokens for LLaVA; (ii) *Mild to Extreme Compression:* retaining $K_{\text{model}} \in \{192, 128, 64, 32, 16\}$ tokens; and (iii) *Blind:* an extreme setting ($K_{\text{model}}$=0) serving as a reference, where no visual tokens are provided to the LLM, so the model responds only based on the language priors.

**Datasets & Evaluation Metric.** We evaluate on three established datasets evaluating diverse capabilities: VQA-v2 (Goyal et al., 2017), TextVQA (Singh et al., 2019), and GQA (Hudson & Manning, 2019). We uniformly sample 1,000 question-image pairs from each dataset for evaluation. To standardize evaluation across free-form LVLM outputs, we employ a uniform answer-matching protocol: following standard text normalization (lowercasing, removing punctuation/articles, and canonicalizing whitespace), a prediction is deemed correct if it matches any ground-truth answer via a whole-word containment check. We report **clean accuracy** and **robust accuracy**, computed as the percentage of correct predictions on clean and adversarial inputs, respectively.

**Baseline.** We adopt VEAttack (Mei et al., 2026), a state-of-the-art encoder-based attack, as our primary baseline. Both VEAttack and our method follow the same gray-box threat model: white-box access to the vision encoder and

black-box access to the downstream LLM. This strict alignment ensures that VEAttack serves as a fair and directly comparable baseline in our evaluation.

**Implementation Setting.** We generate adversarial perturbations using $\ell_{\infty}$-bounded PGD with a budget $\epsilon = 2/255$, step size $\alpha = 0.5/255$, and $T = 100$ iterations with random initialization. Regarding the configuration of CAGE, we utilize the attention weights in the vision encoder as the token importance score $s_i(\mathbf{x})$. To capture budget uncertainty in the EFD component, we employ a uniform prior $P(K_{\text{model}})$ over the range $[16, 192]$ to cover mild-to-extreme compression. The balancing hyperparameter is set to $\lambda = 0.005$. All experiments are conducted on NVIDIA RTX 4090 GPUs.

### 5.2. Main Results

Table 2 reports attack results against five representative token-compression mechanisms on LLaVA. Across all settings, CAGE yields consistently lower robust accuracy than the baseline, indicating that compression-aware optimization exposes stronger vulnerabilities under compressed inference. In addition, our results reveal the following findings.

**Finding 1: CAGE improves attack efficacy across diverse compression mechanisms, including diversity-based selection.** Diversity-driven methods (e.g., DivPrune) often exhibit higher robust accuracy than attention-based mechanisms, as their retained tokens are less aligned with the attention cues primarily exploited by CAGE. Nevertheless, CAGE still yields consistent gains over standard baselines. The gains persist because attention-salient regions are frequently represented within the diverse token set retained by similarity-based selection. Therefore, CAGE can still inject adversarial evidence into the survivor set even under mismatched selection criteria.

**Finding 2: CAGE is particularly effective on text-centric visual understanding tasks.** The relative gain of CAGE is most pronounced on TextVQA, where answers hinge on sparse, highly localized text evidence with limited redundancy, making predictions disproportionately sensitive to the corruption of a few text-bearing survivor tokens under compression. CAGE is designed to concentrate distortion on likely-retained, high-importance tokens and thus more reliably corrupts these decisive text regions than the baseline. As a result, the performance drop is amplified on TextVQA relative to VQA-v2 and GQA, where visual reasoning typically aggregates more distributed and redundant cues.

**Finding 3: CAGE yields consistent advantages across the compression spectrum.** Even without compression (full tokens), CAGE still outperforms the baseline, which we attribute to EFD's token reweighting that focuses perturbation on high-impact visual evidence rather than distributing it uniformly. As the deployment budget decreases from

*Table 2.* Main results comparing clean and robust accuracy (%) across five representative token compression mechanisms on LLaVA. We report **Clean** accuracy and **Robust** accuracy under both the baseline attack (**Base**) and our proposed CAGE. A lower robust accuracy indicates a stronger attack. Green annotations denote the reduction in robust accuracy achieved by CAGE compared to the baseline.

| Compressed Method | GQA | | | TextVQA | | | VQA-v2 | | |
|---|---|---|---|---|---|---|---|---|---|
| | Clean | Robust (Base) | Robust (CAGE) | Clean | Robust (Base) | Robust (CAGE) | Clean | Robust (Base) | Robust (CAGE) |
| *Upper Bound (576 Tokens)* | | | | | | | | | |
| None | 60.3 | 42.3 | $39.4_{(\downarrow 6.9\%)}$ | 57.5 | 34.7 | $26.50_{(\downarrow 23.6\%)}$ | 74.5 | 55.8 | $49.4_{(\downarrow 11.4\%)}$ |
| *Retain 192 Tokens ($K_{model} = 192$)* | | | | | | | | | |
| VisionZIP (CVPR25) | 57.0 | 40.9 | 36.2 | 56.9 | 34.1 | 24.6 | 73.4 | 55.7 | 46.5 |
| VisPruner (ICCV25) | 55.3 | 40.8 | 36.5 | 58.2 | 33.9 | 23.0 | 73.4 | 56.3 | 46.4 |
| DivPrune (CVPR25) | 55.6 | 40.5 | 37.4 | 54.6 | 32.7 | 23.5 | 73.1 | 53.9 | 49.0 |
| FlowCut (NeurIPS25) | 55.4 | 40.3 | 35.1 | 57.8 | 33.6 | 24.3 | 72.9 | 55.9 | 47.3 |
| PruMerge (ICCV25) | 57.2 | 41.5 | 36.0 | 56.0 | 33.3 | 21.6 | 74.2 | 55.1 | 46.8 |
| Average | 56.1 | 40.8 | $36.2_{(\downarrow 11.3\%)}$ | 56.7 | 33.5 | $23.4_{(\downarrow 30.1\%)}$ | 73.4 | 55.4 | $47.2_{(\downarrow 14.8\%)}$ |
| *Retain 128 Tokens ($K_{model} = 128$)* | | | | | | | | | |
| VisionZIP (CVPR25) | 55.1 | 40.9 | 35.6 | 57.0 | 33.5 | 21.6 | 72.1 | 53.8 | 45.1 |
| VisPruner (ICCV25) | 54.3 | 41.5 | 36.2 | 56.5 | 34.1 | 21.5 | 72.2 | 56.3 | 46.8 |
| DivPrune (CVPR25) | 55.6 | 40.4 | 36.5 | 52.7 | 32.2 | 23.2 | 72.2 | 54.0 | 49.2 |
| FlowCut (NeurIPS25) | 54.8 | 40.6 | 34.7 | 57.5 | 33.2 | 21.3 | 70.2 | 54.7 | 44.9 |
| PruMerge (ICCV25) | 55.5 | 41.0 | 35.3 | 55.1 | 32.9 | 21.9 | 72.5 | 55.8 | 47.8 |
| Average | 55.1 | 40.9 | $35.7_{(\downarrow 12.7\%)}$ | 55.8 | 33.2 | $21.9_{(\downarrow 34.0\%)}$ | 71.8 | 54.9 | $46.8_{(\downarrow 14.8\%)}$ |
| *Retain 64 Tokens ($K_{model} = 64$)* | | | | | | | | | |
| VisionZIP (CVPR25) | 52.8 | 40.8 | 34.3 | 54.8 | 32.0 | 18.3 | 69.3 | 53.5 | 43.0 |
| VisPruner (ICCV25) | 53.5 | 40.3 | 34.5 | 54.6 | 31.0 | 18.9 | 70.9 | 53.7 | 45.2 |
| DivPrune (CVPR25) | 52.9 | 40.2 | 36.9 | 50.1 | 29.5 | 21.2 | 71.5 | 54.3 | 48.3 |
| FlowCut (NeurIPS25) | 50.2 | 38.6 | 34.7 | 54.7 | 31.6 | 18.3 | 66.8 | 54.3 | 44.4 |
| PruMerge (ICCV25) | 53.8 | 41.2 | 34.1 | 52.8 | 31.5 | 17.7 | 70.7 | 54.3 | 43.9 |
| Average | 52.6 | 40.2 | $34.9_{(\downarrow 13.2\%)}$ | 53.4 | 31.1 | $18.9_{(\downarrow 39.2\%)}$ | 69.8 | 54.0 | $45.0_{(\downarrow 16.7\%)}$ |
| *Retain 32 Tokens ($K_{model} = 32$)* | | | | | | | | | |
| VisionZIP (CVPR25) | 50.1 | 40.0 | 33.3 | 49.3 | 30.3 | 17.9 | 68.4 | 51.3 | 43.1 |
| VisPruner (ICCV25) | 49.1 | 37.9 | 33.4 | 49.2 | 30.7 | 17.8 | 66.9 | 51.5 | 43.8 |
| DivPrune (CVPR25) | 50.5 | 40.2 | 35.0 | 45.5 | 27.8 | 20.3 | 68.7 | 52.4 | 45.2 |
| FlowCut (NeurIPS25) | 47.2 | 37.2 | 33.9 | 50.1 | 29.3 | 17.4 | 62.2 | 52.6 | 44.1 |
| PruMerge (ICCV25) | 50.4 | 39.7 | 33.9 | 46.1 | 27.6 | 17.5 | 67.6 | 52.8 | 43.6 |
| Average | 49.5 | 39.0 | $33.8_{(\downarrow 13.3\%)}$ | 48.0 | 29.1 | $18.2_{(\downarrow 37.5\%)}$ | 66.8 | 52.1 | $44.0_{(\downarrow 15.5\%)}$ |
| *Retain 16 Tokens ($K_{model} = 16$)* | | | | | | | | | |
| VisionZIP (CVPR25) | 47.4 | 37.2 | 32.4 | 45.8 | 27.2 | 15.8 | 63.2 | 49.7 | 42.7 |
| VisPruner (ICCV25) | 45.9 | 38.9 | 32.8 | 38.2 | 25.1 | 14.7 | 60.4 | 50.9 | 42.4 |
| DivPrune (CVPR25) | 49.3 | 41.2 | 34.1 | 38.5 | 24.3 | 17.0 | 63.8 | 50.1 | 42.9 |
| FlowCut (NeurIPS25) | 40.7 | 37.8 | 33.6 | 38.7 | 25.7 | 15.1 | 55.5 | 50.3 | 44.6 |
| PruMerge (ICCV25) | 47.6 | 38.5 | 32.4 | 39.5 | 25.4 | 15.7 | 65.2 | 50.5 | 41.8 |
| Average | 46.2 | 38.7 | $33.1_{(\downarrow 14.6\%)}$ | 40.1 | 25.5 | $15.7_{(\downarrow 38.4\%)}$ | 61.6 | 50.3 | $42.9_{(\downarrow 14.7\%)}$ |
| *Blind ($K_{model} = 0$)* | | | | | | | | | |
| None | 21.7 | 21.7 | 21.7 | 10.3 | 10.3 | 10.3 | 44.3 | 44.3 | 44.3 |

mild to moderate compression (e.g., 192/128/64 tokens), the advantage typically increases because attack effectiveness increasingly depends on corrupting surviving tokens that CAGE targets explicitly. Under extremely tight budgets (e.g., 32 or 16 tokens), the gain plateaus and may fluctuate on TextVQA and VQA-v2 because robust accuracy approaches the non-visual (blind) reference, leaving little room for further degradation.

### 5.3. Analysis and Ablation

**Impact of Token Compression on Robustness.** We investigate the impact of token compression on robustness from two perspectives: absolute robustness and conditional robustness degradation. First, under a fixed $\ell_\infty$ perturba-

tion budget, the observed robust accuracy $R$ decreases as the token budget shrinks, reflecting the combined effect of information loss and increased susceptibility to adversarial distortion. Second, to disentangle robustness degradation from pure information loss, we introduce conditional robustness degradation (CRD), defined as $\text{CRD} = 1 - \frac{R}{C}$, where $C$ is the clean accuracy. CRD measures the relative fraction of clean performance that is lost under attack, enabling a fairer comparison of attack impact across different compression levels. A higher CRD indicates a stronger attack effect after accounting for the clean accuracy drop caused by compression.

Figure 4 presents the CRD trend under the baseline attack and CAGE. For the baseline attack, CRD generally decreases as the deployment budget shrinks across all three

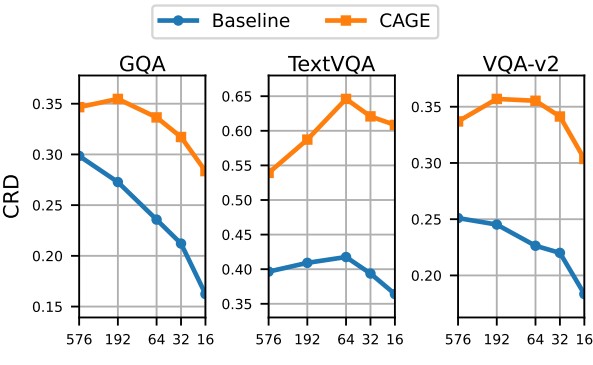

*Figure 4.* **Conditional robustness degradation** (CRD) **vs. deployment token budget.** A higher CRD indicates a larger relative performance drop caused by the attack after accounting for compression-induced clean accuracy degradation. While CRD under the baseline attack generally decreases as the token budget shrinks, CAGE exhibits non-monotonic behavior.

*Table 3.* Defense performance under different dominant token budgets on VQA-v2. Random denotes a baseline that randomly selects additional tokens to examine whether the robustness gain mainly comes from increased randomness/diversity.

| Setting | Full | 192 | 128 | 64 | 32 | 16 |
|---|---|---|---|---|---|---|
| Clean Acc. | 74.5 | 73.4 | 72.1 | 69.3 | 68.4 | 63.2 |
| Robust Acc. | 49.4 | 46.5 | 45.1 | 43.0 | 43.1 | 42.7 |
| Clean Acc. w/ Random | 74.4 | 71.6 | 71.2 | 67.9 | 66.9 | 61.5 |
| Robust Acc. w/ Random | 50.6 | 48.4 | 47.7 | 46.5 | 44.9 | 42.8 |
| Clean Acc. w/ D1 | 74.1 | 72.7 | 70.9 | 68.3 | 66.9 | 61.9 |
| Robust Acc. w/ D1 | 48.2 | 49.9 | 50.0 | 48.4 | 45.3 | 45.5 |
| Clean Acc. w/ D2 | 74.3 | 72.8 | 72.1 | 69.1 | 68.0 | 63.5 |
| Robust Acc. w/ D2 | 50.7 | 48.6 | 47.6 | 46.6 | 46.1 | 42.8 |

*Table 4.* Detection performance using Top-$K$ attention mass as the score on VQA-v2. We report **Acc.** (Detection accuracy), **TPR** (True Positive Rate, measuring attack detection success), **FPR** (False Positive Rate, measuring false alarms on clean inputs), and the **F1** score.

| $K$ | Acc. | TPR | FPR | F1 |
|---|---|---|---|---|
| 1 | 0.900 | 0.850 | 0.050 | 0.895 |
| 2 | 0.920 | 0.920 | 0.080 | 0.920 |
| 4 | 0.940 | 0.910 | 0.030 | 0.938 |
| 16 | 0.860 | 0.910 | 0.190 | 0.867 |

benchmarks, which can misleadingly suggest that stronger compression improves conditional robustness. In contrast, CAGE exhibits a non-monotonic pattern: CRD increases under moderate budgets (e.g., 64–192 tokens) and then partially decreases as the budget shrinks further. This behavior is consistent with compression-aligned optimization. Under moderate compression, clean predictions can still be made from the selected survivor tokens, so clean accuracy $C$ drops only mildly. However, CAGE explicitly targets these same survivors by aligning perturbations with the selection signal, corrupting the evidence that the compressed model relies on. As a result, robust accuracy $R$ decreases more sharply than $C$, which drives CRD upward. Under extremely tight budgets (e.g., 16 tokens), robust accuracy is already severely suppressed, leaving the attack with limited headroom to further reduce $R$, whereas clean accuracy $C$ continues to drop rapidly due to information loss; consequently, CRD decreases.

Appendix E provides further ablations on components, perturbation budgets, hyperparameters, and scoring metrics.

## 6. Possible Defenses

We explore two potential defense strategies against CAGE, detailing the results and analysis on existing defenses in Appendix F.

**Selection-based Defenses.** Since CAGE relies on predicting the deterministic survivor set, we propose two countermeasures: **D1 (Robustness-aware Selection)**, which penalizes tokens with unstable attention scores under random noise; **D2 (Stochastic Candidate Pool)**, which introduces randomness by sampling survivors from a larger candidate

pool. Table 3 presents the defense performance under different deployment budgets with VisionZIP and VQA-v2. The results show that these defenses are somewhat effective, but the gains remain limited. A key reason is an *informativeness-robustness trade-off*: tokens that are stable under perturbations are often weakly informative. For instance, perturbation-insensitive patches (e.g., backgrounds or low-texture regions) typically lack the semantic density needed for downstream reasoning. As a result, explicitly prioritizing such robust (or random) tokens tends to discard task-critical visual evidence, substantially degrading clean performance and ultimately diminishing the practical utility.

**Attention-based Detection.** We observe that adversarial attacks tend to disperse the attention distribution. Leveraging this finding, we propose a detector based on the Top-$K$ attention mass. Specifically, we compute the cumulative attention assigned to the $K$ highest-attention tokens and use it as a detection score to distinguish adversarial inputs. Results are reported in Table 4, where this simple detector achieves a detection accuracy of 0.94 with $K=4$. However, this detector relies on a threshold tuned on the attacks we evaluate. When the attacker uses a different attack strategy, the attention statistics may shift and the same threshold may no longer work well (see Appendix F). Overall, attention-based detection shows promise but necessitates more robust designs that combine multiple statistics.

# 7. Conclusion

In this paper, we present the first study on the adversarial robustness of LVLMs under visual token compression. We identify a critical optimization-inference mismatch in existing attacks, which causes overestimating the robustness of compressed LVLMs. To bridge this gap, we propose CAGE, an adversarial attack that aligns adversarial optimization with the compression bottleneck. Extensive experiments demonstrate that CAGE significantly outperforms the SOTA baseline under different deployment token budgets. We further explore potential defenses and find that they offer partial mitigation but remain insufficient. This work serves as a wake-up call for the community to incorporate security evaluations and defenses in the design of efficient LVLMs.

# Acknowledgements

This work was supported by the Ministry of Science and Technology of the People's Republic of China (National Key Research and Development Programme, Grant No: 2025YFE0200100), the National Natural Science Foundation of China (Grant No: 62502416), the Research Grants Council (Grant No: 15209922 and 15207725), Hong Kong SAR, China.

# Impact Statement

This work studies adversarial robustness of large vision-language models (LVLMs) under visual token compression. Since we develop and evaluate attacks, our findings could be misused to degrade deployed systems. We therefore present CAGE as a diagnostic tool: it reveals failure modes introduced by compression and enables more faithful robustness evaluation. To reduce misuse risk, we do not release any automated pipeline that would directly facilitate real-world abuse. Instead, we focus on controlled experiments, principled analysis, and actionable measurements that support reproducible auditing and defense development. We also explore several potential defenses as initial mitigation steps. Crucially, this study highlights the overlooked security implications of token compression. By establishing a rigorous evaluation protocol, we aim to shift the community's focus from purely efficiency-driven designs to mechanisms that are inherently robust against adversarial manipulation.

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

# A. Algorithm

---

**Algorithm 1** CAGE

---

**Input:** Vision encoder $f_\theta$, clean image $\mathbf{x}$.
**Input:** PGD parameters: step size $\alpha$, budget $\epsilon$, iterations $T$.
**Input:** CAGE parameters: budget prior $P(K)$, hyperparameter $\lambda$.
**Output:** Adversarial perturbation $\boldsymbol{\delta}$.

1: Initialize $\boldsymbol{\delta}^{(0)} \leftarrow \mathbf{0}$ (or random initialization within $\epsilon$-ball).
2: **for** $t = 0$ to $T - 1$ **do**
3:     $\mathbf{x}_{adv}^{(t)} \leftarrow \mathbf{x} + \boldsymbol{\delta}^{(t)}$
4:     // Forward pass to get features and scores
5:     Get clean features $\mathbf{z}^{\text{cln}}$, adversarial features $\mathbf{z}^{\text{adv}}$, and attention scores $s(\mathbf{x}_{adv}^{(t)})$ from $f_\theta$.
6:     // — Component I: Expected Feature Disruption (EFD) —
7:     Compute per-token distortion: $d_i \leftarrow 1 - \cos(\mathbf{z}_i^{\text{adv}}, \mathbf{z}_i^{\text{cln}})$.
8:     Compute current ranks $r_i$ based on scores $s_i$.
9:     Compute survival probabilities: $\pi_i \leftarrow \sum_k P(K = k) \cdot \mathbb{I}(k > r_i)$.
10:    $\mathcal{L}_{\text{EFD}} \leftarrow \frac{\sum \pi_i d_i}{\sum \pi_i}$.
11:    // — Component II: Rank Distortion Alignment (RDA) —
12:    Compute distortion distribution: $p_d(i) \leftarrow \text{softmax}(d_i)$, with gradients detached.
13:    Compute selection distribution: $p_s(i) \leftarrow \text{softmax}(s_i)$.
14:    $\mathcal{L}_{\text{RDA}} \leftarrow \sum p_d(i) \cdot \log p_s(i)$.
15:    // — Joint Optimization (Gradient Ascent) —
16:    $\mathcal{L}_{\text{total}} \leftarrow \mathcal{L}_{\text{EFD}} + \lambda \cdot \mathcal{L}_{\text{RDA}}$.
17:    Compute gradient: $\mathbf{g}^{(t)} \leftarrow \nabla_{\boldsymbol{\delta}} \mathcal{L}_{\text{total}}(\mathbf{x}_{adv}^{(t)})$.
18:    Update perturbation: $\boldsymbol{\delta}^{(t+1)} \leftarrow \text{Proj}_\epsilon \left( \boldsymbol{\delta}^{(t)} + \alpha \cdot \text{sign}(\mathbf{g}^{(t)}) \right)$.
19: **end for**
    **return** $\boldsymbol{\delta} \leftarrow \boldsymbol{\delta}^{(T)}$.

---

# B. Related Work

**LVLM Efficiency.** The integration of visual inputs into LLMs leads to long token sequences and substantial computational and memory overhead (Liu et al., 2024a). Such overhead becomes especially restrictive in latency-sensitive scenarios, including LVLM-driven mobile agents and reasoning-intensive applications, where efficient inference is critical (Yao et al., 2025a;b). This has spurred growing interest in visual token compression for LVLMs. Existing token compression approaches can be categorized by *where* token reduction is applied relative to the language model, as shown in Table 5. ① *Inner-LLM* approaches (Chen et al., 2024a; Zhang et al., 2025d; Yin et al., 2025; Liu et al., 2024b; Shao et al., 2025), such as FastV (Chen et al., 2024a) and SparseVLM (Zhang et al., 2025d), integrate token compression into the language model's transformer layers, reducing the effective number of visual tokens processed during decoding. ② *Outer-LLM* approaches perform token selection or aggregation *before* the main language model computation, treating the LLM as a black box. Representative methods include PruMerge (Shang et al., 2025), VisionZip (Yang et al., 2025), VisPruner (Zhang et al., 2025b), and FlowCut (Tong et al., 2025), which differ mainly in how token importance or redundancy is estimated (e.g., attention-based scoring, similarity-based pruning, or cross-layer information flow), but share a common paradigm of producing a compact visual token sequence prior to LLM inference. In this work, we focus on *outer-LLM* compression methods, as they are plug-and-play, require no modification to the language model, and are broadly compatible with existing inference and acceleration frameworks.

**LVLM Robustness.** The security and privacy problems of deep learning models have received growing attention (Zhang et al., 2025c; Bai et al., 2025a;b). In particular, the robustness of LVLMs has emerged as a critical concern as these systems are increasingly deployed in real-world applications, such as medical image analysis (Nath et al., 2025; Lin et al., 2025) and autonomous systems (Lübberstedt et al., 2025). Existing adversarial attacks on LVLMs broadly fall into two optimization paradigms. ① End-to-end attacks backpropagate through the entire multimodal pipeline to craft adversarial images, but are often computationally expensive due to large models and long contexts (Schlarmann & Hein, 2023). ② Encoder-based attacks optimize perturbations by targeting only the vision encoder, providing a significantly lighter alternative (Zhao

*Table 5.* Token compression mechanisms. "A" and "S" denote attention- and similarity-based metrics, respectively.

| Method | Venue | Outer-LLM | | Inner-LLM | |
|---|---|---|---|---|---|
| | | Prune | Merge | Prune | Merge |
| FastV (Chen et al., 2024a) | ECCV 2024 | ✗ | ✗ | A | ✗ |
| SparseVLM (Zhang et al., 2025d) | ICML 2025 | ✗ | ✗ | A | S |
| DART (Yin et al., 2025) | arXiv 2025 | ✗ | ✗ | S | ✗ |
| MustDrop (Liu et al., 2024b) | arXiv 2024 | A | S | A | ✗ |
| HoliTom (Shao et al., 2025) | arXiv 2025 | ✗ | S | ✗ | S |
| VisionZip (Yang et al., 2025) | CVPR 2025 | A | S | ✗ | ✗ |
| VisPruner (Zhang et al., 2025b) | ICCV 2025 | A+S | ✗ | ✗ | ✗ |
| DivPrune (Alvar et al., 2025) | CVPR 2025 | S | ✗ | ✗ | ✗ |
| FlowCut (Tong et al., 2025) | NeurIPS 2025 | A+S | ✗ | ✗ | ✗ |
| PruMerge (Shang et al., 2025) | ICCV 2025 | A | S | ✗ | ✗ |
| G-Prune (Jiang et al., 2025) | AAAI 2025 | S | ✗ | ✗ | ✗ |
| HiRED (Arif et al., 2025) | AAAI 2025 | A | ✗ | ✗ | ✗ |

et al., 2023a; Dong et al., 2023; Cui et al., 2024; Wang et al., 2024c; Xie et al., 2025; Zhang et al., 2026). For instance, VT-Attack (Wang et al., 2024c) perturbs encoded visual tokens to break the vision encoder's token representations. For prompt-diverse tasks such as VQA, VEAttack (Mei et al., 2026) crafts downstream-agnostic examples by minimizing the cosine similarity between clean and perturbed visual token features, and further provides theoretical analysis establishing a lower-bound misalignment guarantee in the LLM-aligned representation space. However, existing encoder-based attacks are typically optimized on full-token models, while token compression reshapes the encoder's token space, motivating our study of robustness in recently compressed LVLMs.

## C. Detailed Experimantal Setup

### C.1. Datasets

We evaluate on diverse datasets that cover complementary domains essential to real-world LVLM applications.

**VQA-v2** (Goyal et al., 2017) is a large-scale open-ended visual question answering benchmark built on COCO images, where each question is annotated with multiple human answers. It covers a broad spectrum of everyday visual concepts (objects, attributes, actions, counting, and commonsense cues), and is widely used as a representative testbed for general-purpose VQA. In our study, VQA-v2 serves as a *general perception and language grounding* benchmark, reflecting the typical deployment scenario of LVLMs for open-domain visual understanding.

**TextVQA** (Singh et al., 2019) is designed for *scene-text-centric* reasoning: questions often require reading and understanding text embedded in natural images (e.g., signs, product packages, screens, menus), and then integrating it with visual context. This benchmark stresses the model's OCR-related capability and fine-grained visual grounding under cluttered backgrounds. We include TextVQA because token compression may disproportionately affect small, high-frequency visual cues (such as characters and short words), and thus its robustness behavior can differ substantially from VQA.

**GQA** (Hudson & Manning, 2019) focuses on *compositional visual reasoning*, featuring structured questions that frequently require multi-hop reasoning over objects, relations, and attributes (e.g., spatial relations, comparisons, logical conjunctions). Compared to open-ended VQA, GQA emphasizes systematic generalization and relational grounding. We use GQA to test whether compression-aligned attacks remain effective when model decisions rely more heavily on relational evidence and compositional structure rather than isolated salient cues.

### C.2. Compression Mechanism

We briefly summarize the visual token compression methods evaluated in this paper.

**VisionZip (Yang et al., 2025)** adopts a *select-then-merge* pipeline focused on redundancy reduction. It first ranks visual tokens based on importance scores (e.g., attention) and retains the Top-$K$ tokens as survivors. To minimize information loss, the discarded tokens are merged into the semantically similar survivors, thereby reducing the sequence length while preserving relevant visual details. In our experiments, we set the number of merged tokens to 10% of the whole token budget.

*Table 6.* Attack performance (%) of various compressed methods on Qwen2.5-VL across different datasets. We report clean accuracy (Clean) and robust accuracy under a baseline attack (Adv (Base)) and our compression-aligned attack (Adv (CAGE)); lower robust accuracy indicates a stronger attack. Green annotations indicate the relative decrease of Adv (CAGE) over Adv (Base).

| Compressed Method | GQA | | | TextVQA | | | VQA-v2 | | |
|---|---|---|---|---|---|---|---|---|---|
| | Clean | Adv (Base) | Adv (CAGE) | Clean | Adv (Base) | Adv (CAGE) | Clean | Adv (Base) | Adv (CAGE) |
| *Upper Bound (144 Tokens)* | | | | | | | | | |
| None | 54.4 | 35.5 | 35.2$_{(\downarrow 0.8\%)}$ | 70.3 | 38.0 | 33.9$_{(\downarrow 10.8\%)}$ | 65.5 | 40.3 | 40.2$_{(\downarrow 0.2\%)}$ |
| *Retain 72 Tokens ($K_{model} = 72$)* | | | | | | | | | |
| VisionZIP (CVPR25) | 54.0 | 33.1 | 33.0 | 66.2 | 29.6 | 24.2 | 57.4 | 35.7 | 28.1 |
| VisPruner (ICCV25) | 52.5 | 36.1 | 36.1 | 62.8 | 29.4 | 26.0 | 54.9 | 36.0 | 33.7 |
| DivPrune (CVPR25) | 53.3 | 36.4 | 35.1 | 58.4 | 34.0 | 33.8 | 58.4 | 38.9 | 35.1 |
| **Average** | 53.3 | 35.2 | 34.7$_{(\downarrow 1.3\%)}$ | 62.5 | 31.0 | 28.0$_{(\downarrow 9.7\%)}$ | 56.9 | 36.9 | 32.3$_{(\downarrow 12.4\%)}$ |
| *Retain 36 Tokens ($K_{model} = 36$)* | | | | | | | | | |
| VisionZIP (CVPR25) | 47.9 | 32.3 | 30.1 | 56.1 | 19.2 | 13.0 | 53.6 | 28.4 | 27.5 |
| VisPruner (ICCV25) | 47.0 | 34.4 | 33.4 | 42.4 | 19.6 | 17.5 | 50.5 | 33.1 | 31.7 |
| DivPrune (CVPR25) | 51.1 | 36.5 | 36.0 | 57.7 | 28.6 | 24.1 | 55.3 | 36.2 | 34.9 |
| **Average** | 48.7 | 34.4 | 33.2$_{(\downarrow 3.5\%)}$ | 52.1 | 22.5 | 18.2$_{(\downarrow 19.1\%)}$ | 53.1 | 32.6 | 31.4$_{(\downarrow 3.7\%)}$ |
| *Blind ($K_{model} = 0$)* | | | | | | | | | |
| None | 28.4 | 28.4 | 28.4 | 6.8 | 6.8 | 6.8 | 17.0 | 17.0 | 17.0 |

**VisPruner (Zhang et al., 2025b)** performs *saliency-based pruning*. It utilizes vision-encoder attention maps to estimate token importance, identifying and retaining a compact subset of highly informative tokens. The method explicitly filters out background redundancy to maximize the semantic density of the pruned sequence under a fixed budget.

**DivPrune (Alvar et al., 2025)** is a *diversity-driven* selection mechanism. Unlike standard methods that prioritize high-activation regions, DivPrune selects a subset of tokens that maximizes feature diversity. This strategy reduces redundancy among retained tokens and ensures broad coverage of distinct image regions, preventing the model from over-focusing on a single salient object.

**FlowCut (Tong et al., 2025)** assesses token redundancy through *attention flow* rather than static importance scores. It analyzes cross-layer token interactions to estimate the information flow, pruning tokens that contribute minimally to the global context. This allows for a more structural reduction of redundancy based on the network's internal information propagation.

**PruMerge (Shang et al., 2025)** employs a hybrid *pruning-and-merging* strategy. It first selects a set of representative tokens based on importance and then merges the remaining tokens into these representatives (e.g., via weighted averaging based on similarity). This approach balances the efficiency of pruning with the information preservation of merging.

# D. Attack Performance on Other LVLMs

## D.1. Qwen2.5-VL

Table 6 reports the attack performance on Qwen2.5-VL under different compression settings. To ensure that the token budget is computed accurately, we resize all input images to $336 \times 336$, yielding a fixed number of visual tokens in Qwen. We verify attack performance under three representative compression mechanisms, since most existing methods do not release Qwen-compatible implementations and their official codebases are tightly coupled to specific LLaVA. Overall, our attack (CAGE) consistently achieves lower robust accuracy than the baseline attack across all datasets and token budgets, indicating stronger attack effectiveness. However, compared to the results on LLaVA, the relative improvement brought by CAGE on Qwen2.5-VL is moderate.

This observation can be attributed to the architectural characteristics of Qwen2.5-VL. Since the visual encoder of Qwen natively integrates token merging, the baseline attack is optimized directly within this compressed feature space. This minimizes the optimization-inference mismatch that typically affects models like LLaVA. Consequently, the performance gain of our method appears less pronounced on Qwen2.5-VL than on models using dense patch sets. Despite this built-in compression, CAGE still demonstrates clear advantages, especially under more aggressive token budgets. For instance, CAGE achieves average relative decreases reaching up to 19.1% on TextVQA and $K_{model} = 36$. This indicates that explicitly

*Table 7.* Attack performance (%) on InternVL3 and VQA-v2 under different visual token compression settings. We report clean accuracy (Clean) and robust accuracy under a baseline attack (Adv (Base)) and our compression-aligned attack (Adv (CAGE)); lower robust accuracy indicates a stronger attack. Green annotations indicate the relative decrease of Adv (CAGE) over Adv (Base).

| Compressed Method | Clean | Adv (Base) | Adv (CAGE) |
|---|---|---|---|
| *Upper Bound (144 Tokens)* | | | |
| None | 71.1 | 43.6 | $40.0_{(\downarrow 8.3\%)}$ |
| *Retain 72 Tokens ($K_{model} = 72$)* | | | |
| VisionZIP (CVPR25) | 70.8 | 39.9 | 33.4 |
| VisPruner (ICCV25) | 69.7 | 40.8 | 35.9 |
| DivPrune (CVPR25) | 69.5 | 41.6 | 38.1 |
| **Average** | 70.0 | 40.8 | $35.8_{(\downarrow 12.3\%)}$ |
| *Retain 36 Tokens ($K_{model} = 36$)* | | | |
| VisionZIP (CVPR25) | 66.0 | 35.5 | 30.8 |
| VisPruner (ICCV25) | 66.0 | 36.7 | 33.3 |
| DivPrune (CVPR25) | 66.3 | 38.3 | 34.4 |
| **Average** | 66.1 | 36.8 | $32.8_{(\downarrow 10.9\%)}$ |

*Table 8.* Robust accuracy (%) comparing the EFD-only variant and the complete CAGE design under different budgets on VQA-v2.

| Method | 576 (Full) | 192 | 128 | 64 | 32 | 16 |
|---|---|---|---|---|---|---|
| EFD Only | 50.8 | 48.5 | 47.0 | 45.8 | 46.0 | 45.7 |
| CAGE | 49.4 | 46.5 | 45.1 | 43.0 | 43.1 | 42.7 |

modeling the interaction between token selection and adversarial perturbations remains beneficial, even for LVLMs that already employ internal token merging.

### D.2. InternVL3-8B

Table 7 further reports the attack performance on InternVL3-8B under different visual token compression settings on VQA-v2. This experiment verifies whether the effectiveness of CAGE generalizes beyond LLaVA-style architectures and Qwen2.5-VL. Overall, CAGE consistently achieves lower robust accuracy than the baseline attack across all compression methods and token budgets, demonstrating that compression-aligned perturbation optimization remains effective on InternVL3-8B. For example, under VisionZIP with $K_{model} = 72$, CAGE reduces the robust accuracy from 39.9% to 33.4%, corresponding to a relative decrease of 16.3%. Under more aggressive compression with $K_{model} = 36$, CAGE also consistently improves attack effectiveness across VisionZIP, VisPruner, and DivPrune.

These results suggest that the optimization-inference mismatch is not limited to a specific LVLM backbone or compression algorithm. Although InternVL3-8B adopts a different visual representation and multimodal alignment architecture from LLaVA and Qwen2.5-VL, explicitly modeling the interaction between token compression and adversarial perturbations still provides clear benefits. This further supports the generality of our compression-aligned attack design.

## E. Additional Ablation and Analysis

### E.1. Component Ablation

We conduct ablations on CAGE to isolate the impact of each design component on attack performance. Table 8 reports results on VQA-v2 for LLaVA under VisionZip, comparing an EFD-only variant against the full CAGE across token budgets. Across token budgets, CAGE consistently achieves lower robust accuracy than the EFD-only variant, indicating that adding RDA provides a clear benefit. This result suggests that reweighting distortion alone (EFD) is insufficient: by explicitly steering the selection signal, RDA increases the likelihood that highly perturbed tokens are retained after compression, allowing adversarial evidence to pass through the bottleneck and further reducing robustness.

*Table 9.* Robust accuracy (%) under different token budgets on VQA-v2.

| $K_{model}$ | Budget = 2 / 255 | | Budget = 4 / 255 | | Budget = 8 / 255 | |
|:---:|:---:|:---:|:---:|:---:|:---:|:---:|
| | Baseline | CAGE | Baseline | CAGE | Baseline | CAGE |
| 576 (Full) | 55.8 | 49.4 | 47.1 | 45.5 | 46.2 | 45.2 |
| 192 | 55.7 | 46.5 | 46.1 | 44.9 | 42.5 | 42.6 |
| 128 | 53.8 | 45.1 | 46.2 | 43.5 | 43.9 | 42.9 |
| 64 | 53.5 | 43.0 | 46.1 | 41.2 | 43.2 | 42.2 |
| 32 | 51.3 | 43.1 | 45.8 | 42.1 | 43.1 | 42.0 |
| 16 | 49.7 | 42.7 | 44.7 | 41.1 | 43.2 | 38.9 |

*Table 10.* Impact of hyperparameter $\lambda$ on attack effectiveness under different dominant token budgets on VQA-v2. We report robust accuracy (%), where lower values indicate stronger attacks.

| $K_{model}$ | $\lambda = 0$ (EFD only) | $\lambda = 0.1$ | $\lambda = 0.05$ | $\lambda = 0.01$ | $\lambda = 0.005$ | $\lambda = 0.001$ |
|:---:|:---:|:---:|:---:|:---:|:---:|:---:|
| 576 (Full) | 50.8 | 51.8 | 49.5 | 50.5 | **49.4** | 50.1 |
| 192 | 48.5 | 49.0 | **46.5** | 46.6 | **46.5** | 47.7 |
| 128 | 47.0 | 46.0 | 45.3 | **44.9** | 45.1 | 45.3 |
| 64 | 45.8 | 45.4 | 45.9 | 43.4 | **43.0** | 43.8 |
| 32 | 46.0 | 44.8 | 44.6 | **42.9** | 43.1 | 43.0 |
| 16 | 45.7 | 43.0 | 43.6 | 43.0 | **42.7** | 43.6 |

### E.2. Different Perturbation Budgets

We further study how attack effectiveness scales with the perturbation budget. Table 9 compares the attack performance under different perturbation budgets $\epsilon \in \{2/255, 4/255, 8/255\}$. The results show that CAGE consistently outperforms the baseline across all perturbation budgets, with the advantage becoming more pronounced under smaller token budgets. In particular, in the low-budget regime ($\epsilon$=2/255), CAGE provides the largest gains because the perturbation budget is scarce and must be allocated efficiently: the baseline dilutes optimization over tokens that will later be pruned, while CAGE concentrates distortion on the likely-retained survivor tokens, yielding substantially stronger attacks (e.g., 49.7%→42.7% at 16 tokens). As $\epsilon$ increases, both attacks become stronger and robust accuracy decreases overall, confirming the expected monotonic trend with larger perturbation budgets. Meanwhile, the relative gap may narrow at higher $\epsilon$ due to diminishing returns: with sufficient budget, the baseline can partially compensate for dilution by injecting stronger distortion, and robust accuracy approaches a performance floor with limited remaining headroom for further degradation.

### E.3. The Impact of $\lambda$

We analyze the sensitivity of CAGE to the weighting coefficient $\lambda$, which balances the two objectives. Results are reported in Table 10. Overall, moderate values consistently perform best: $\lambda$=0.005 yields the lowest accuracy for most budgets (Full/64/16), while $\lambda$=0.01 is slightly better in a few cases (128/32) and remains competitive across the board. Importantly, most non-zero $\lambda$ settings are comparable to or better than $\lambda$=0, indicating that incorporating RDA is generally beneficial even without delicate tuning. In contrast, overly large $\lambda$ (e.g., 0.1) generally weakens the attack, suggesting that over-emphasizing the auxiliary alignment term can distract optimization from inducing sufficient feature disruption. Meanwhile, extremely small $\lambda$ (e.g., 0.001) also degrades performance, indicating that without enough alignment pressure, adversarial distortion is less likely to survive the compression bottleneck. These results imply that a modest $\lambda$ is crucial to jointly achieve strong feature disruption and effective bottleneck alignment, and we adopt $\lambda$=0.005 as a robust default.

### E.4. Impact of Token Importance Metric

We study the impact of different token scoring metrics used in the attack pipeline to determine which signal better guides the adversarial optimization. Specifically, we compare two representative metrics:

- $\ell_1$-**norm score:** Prior work (Tong et al., 2025; Guo et al., 2024) shows that the $\ell_1$ norm of each token's value vector is a proxy for information strength, which can measure the token importance. This metric ranks tokens by embedding

*Table 11.* Impact of different score metrics on attack effectiveness. We report robust accuracy (%), where lower values indicate stronger attacks.

| $K_{\mathrm{model}}$ | Norm | Attention |
|---|---|---|
| 576 (Full) | 65.2 | **49.4** |
| 192 | 65.2 | **46.5** |
| 128 | 62.5 | **45.1** |
| 64 | 60.6 | **43.0** |
| 32 | 56.3 | **43.1** |
| 16 | 51.0 | **42.7** |

*Table 12.* Impact of $K_{\mathrm{min}}$ and $K_{\mathrm{max}}$ on attack effectiveness under different dominant token budgets on VQA-v2. We report robust accuracy (%), where lower values indicate stronger attacks.

| $K_{\mathrm{model}}$ | $[16, 64]$ | $[16, 128]$ | $[16, 192]$ | $[16, 384]$ |
|---|---|---|---|---|
| 576 (Full) | 54.2 | 50.2 | 49.4 | **49.0** |
| 192 | 52.4 | 48.1 | 46.5 | **46.2** |
| 128 | 50.0 | 46.9 | **45.1** | 46.5 |
| 64 | 45.6 | 44.3 | **43.0** | 45.6 |
| 32 | **42.0** | 45.4 | 43.1 | 45.8 |
| 16 | **42.3** | 43.3 | 42.7 | 44.7 |

magnitude, based on the intuition that smaller norms correspond to weaker signals and thus convey less visual information through the Value vectors.

- **Attention score:** We rank tokens by their attention-based importance, which reflects each token's contribution to multimodal reasoning. Compared to norm-based scoring, attention better captures semantic relevance to the textual context. Attention-based importance has been widely adopted in token selection and compression for LVLMs (Yang et al., 2025; Tong et al., 2025; Arif et al., 2025).

As shown in Table 11, the attention-based metric consistently leads to significantly lower robust accuracy than the norm-based alternative across all dominant token budgets. This indicates that attention provides a more effective signal for identifying tokens whose perturbation has a larger impact on the model output. This behavior can be explained by the semantic nature of attention weights in LVLMs. While embedding norms mainly capture low-level activation magnitude, they do not necessarily correlate with a token's influence on cross-modal alignment or downstream reasoning. In contrast, attention scores explicitly encode how strongly visual tokens interact with language tokens during inference, making them a more reliable proxy for token importance in multimodal decision making.

### E.5. The Impact of budget-prior range $[K_{\mathbf{min}}, K_{\mathbf{max}}]$

Table 12 examines how the budget-prior range $[K_{\mathrm{min}}, K_{\mathrm{max}}]$ affects attack strength. Overall, we observe a clear trade-off between *perturbation concentration* and *attack coverage*. With a narrow prior (e.g., $[16, 64]$), CAGE concentrates optimization on only the top-ranked tokens, yielding the strongest attacks under aggressive compression where a handful of survivors dominate inference (e.g., $42.0\%$ at budget $= 32$). However, this choice is less effective at higher budgets (e.g., $54.2\%$ at Full), since many tail/context tokens remain weakly perturbed, allowing the model to recover using relatively clean complementary evidence. In contrast, widening the prior (e.g., $[16, 384]$) spreads perturbations over a larger portion of the token set, improving effectiveness against high-budget models (reducing Full accuracy to $49.0\%$), but introducing budget dilution that weakens the distortion concentrated on the most influential survivors under extreme compression (e.g., $44.7\%$ at budget $= 16$). Among the tested settings, $[16, 192]$ provides the best overall balance: it preserves strong disruption on dominant survivors in low-budget regimes while maintaining sufficient coverage to degrade performance in high-budget settings, making it a robust default when the deployment budget is unknown. A practical rule is to set $K_{\mathrm{min}}$ to the most aggressive budget you want to be robust against (smallest survivors) and set $K_{\mathrm{max}}$ to the largest budget expected in deployment (largest survivors), while avoiding overly large $K_{\mathrm{max}}$ that causes dilution.

*Table 13.* Attack performance on more baselines.

| Method | 576 (Full) | 192 | 128 | 64 | 32 | 16 |
|---|---|---|---|---|---|---|
| Clean | 74.5 | 73.4 | 72.1 | 69.3 | 68.4 | 63.2 |
| AttackVLM-ii (Zhao et al., 2023b) | 62.6 | 62.1 | 62.1 | 60.8 | 57.3 | 52.5 |
| VT-Attack (Wang et al., 2024b) | 59.4 | 60.2 | 60.1 | 59.2 | 54.7 | 51.1 |
| VEAttack (Mei et al., 2026) | 55.8 | 55.7 | 53.8 | 53.5 | 51.3 | 49.7 |
| CAGE | 49.4 | 46.5 | 45.1 | 43.0 | 43.1 | 42.7 |

*Table 14.* Attack performance on image captioning task.

| Method | 576 (Full) | 192 | 128 | 64 | 32 | 16 |
|---|---|---|---|---|---|---|
| Clean | 1.2764 | 1.2941 | 1.3014 | 1.1757 | 1.0075 | 0.6941 |
| Baseline | 0.2202 | 0.2261 | 0.2317 | 0.1998 | 0.1689 | 0.1199 |
| CAGE | 0.1699 | 0.1740 | 0.1285 | 0.1090 | 0.1166 | 0.0850 |

### E.6. Comparing with More Baselines

In the main paper, we use VEAattack as the primary baseline, since it is the strongest and most relevant visual attack baseline in our setting. To further verify that our conclusions do not depend on a single baseline choice, we additionally compare with two representative attacks, AttackVLM-ii (Zhao et al., 2023b) and VT-Attack (Wang et al., 2024b), on VisionZip and VQA-v2.

Table 13 reports the adversarial accuracy under different token budgets. Across all compression levels, our method consistently achieves lower adversarial accuracy than all competing baselines, including the full-token setting. This shows that the advantage of CAGE is not limited to a comparison against VEAattack alone, but generalizes to multiple representative LVLM attack baselines. The gains are particularly clear under compressed settings, which is consistent with our main claim that compression-aware attack design is more effective against compressed LVLMs.

### E.7. Attack Performance on Image Captioning

We further evaluate the attack performance on the image captioning task. Following the same compressed-LVLM setting, we conduct experiments under VisionZip with different token budgets and report results on 100 samples from the COCO dataset. We use CIDEr as the evaluation metric, where lower scores under attack indicate stronger degradation of caption quality.

As shown in Table 14, CAGE consistently achieves lower CIDEr scores than the baseline across all token budgets, demonstrating stronger attack effectiveness not only on VQA but also on generative captioning tasks. The advantage is particularly clear under moderate and strong compression (e.g., 128/64/32 tokens), where compression-aware attack design more effectively disrupts the visual evidence retained by the compressed model. These results suggest that the benefit of CAGE generalizes in different tasks.

### E.8. Runtime Comparison

Table 15 compares the runtime of different attack methods. CAGE requires 5.74 seconds per sample, corresponding to only $1.07\times$ the cost of VEAattack, while remaining substantially more efficient than VT-Attack (9.63 seconds, $1.80\times$). This suggests that the stronger attack performance of CAGE comes with only marginal additional computational overhead.

### E.9. Black-box Setting

We further study a fully black-box transfer setting by using a surrogate vision encoder different from that of the target compressed LVLM. Specifically, attacks are optimized on the surrogate encoder openai/clip-vit-base-patch16, while evaluation is performed on the target compressed LVLM under VisionZip with different token budgets. For both the baseline and CAGE, we use the same perturbation budget and optimization protocol: $\epsilon = 16/255$, step size $\alpha = 1/255$, and 100 PGD steps.

*Table 15.* Runtime comparison of different attack methods. We report the average runtime per sample, together with the relative runtime normalized to VEAattack. Lower is better.

| Method | Avg. time / sample (s) | Relative cost vs. VEAattack |
|---|---|---|
| VEAattack | 5.35 | 1.00× |
| AttackVLM-ii | 5.28 | 0.99× |
| VT-Attack | 9.63 | 1.80× |
| CAGE | 5.74 | 1.07× |

*Table 16.* Attack performance on other encoder.

| Method | 576 (Full) | 192 | 128 | 64 | 32 | 16 |
|---|---|---|---|---|---|---|
| Clean | 74.5 | 73.4 | 72.1 | 69.3 | 68.4 | 63.2 |
| Baseline | 69.5 | 69.4 | 68.7 | 65.5 | 64.4 | 63.0 |
| CAGE | 69.4 | 69.1 | 66.9 | 64.8 | 63.1 | 62.2 |

The results are shown in Table 16. Compared with the gray-box setting in the main paper, the additional gain of CAGE over the baseline becomes more limited in this fully black-box setting, although CAGE still achieves lower adversarial accuracy under most compressed budgets. We believe this behavior is expected. In the fully black-box case, the attacker not only needs the perturbation itself to transfer across encoders, but also needs the *compression-aware signal* to transfer, i.e., the perturbation should remain aligned with the token-selection and retention behavior of the target compressed LVLM. This is substantially more challenging, because such signals are strongly coupled with the target model's visual encoder and compression pipeline.

These results suggest that the main advantage of CAGE is strongest in the gray-box regime, where the attacker knows or can closely approximate the victim model's vision encoder and compression mechanism. By contrast, when the visual encoder is entirely unknown, obtaining useful compression-aware signals for the target model becomes difficult, which limits the extra gain over a generic baseline. Importantly, this does not contradict our main claim; rather, it highlights that beyond standard black-box transferability, exploiting the compression-induced bottleneck itself is an additional challenge in the fully black-box setting.

### E.10. Additional Analysis on Budget Uncertainty and Optimizers

We further evaluate whether the improvement of CAGE can be reproduced by simpler budget sampling or stronger optimizers. First, we compare CAGE with an EOT-style baseline that samples multiple candidate token budgets at each iteration and optimizes the averaged loss over the corresponding retained tokens. Second, we test CAGE with APGD and MI-FGSM to examine whether the gain depends on the specific optimizer. The results are reported in Table 17 and Table 18.

**Comparison with EOT.** A natural alternative for handling unknown deployment budgets is to use an EOT-style objective, where the attack samples multiple candidate token budgets during optimization and maximizes the average loss. As shown in Table 17, CAGE consistently achieves lower robust accuracy than the EOT baseline across all evaluated token budgets. For example, when $K_{\text{model}} = 64$, CAGE reduces the robust accuracy from 49.8% to 43.0%. Similarly, at $K_{\text{model}} = 16$, CAGE further reduces robust accuracy from 47.6% to 42.7%. These results suggest that simply averaging over sampled budgets is insufficient. Unlike EOT, CAGE explicitly models token survival probabilities and further aligns token distortion with the selection ranking, leading to a more structured compression-aware attack objective.

**Effect of optimizers.** We also evaluate whether the advantage of CAGE depends on the use of PGD. As shown in Table 18, CAGE consistently outperforms VEAttack under APGD, MI-FGSM, and PGD. This indicates that the improvement mainly comes from the compression-aligned objective rather than from a particular optimizer. Interestingly, stronger or more recent optimizers do not necessarily lead to better attack performance in our setting. This is likely because these optimizers are mostly designed for standard classification objectives, whereas our attack optimizes feature-level disruption in LVLM visual representations. Therefore, directly transferring these optimizers to the compressed-LVLM setting does not always provide additional benefits.

*Table 17.* Attack performance (%) of the EOT-style budget-sampling baseline and CAGE on VQA-v2. Lower robust accuracy indicates a stronger attack.

| Method | 576 | 192 | 128 | 64 | 32 | 16 |
|--------|-----|-----|-----|----|----|----|
| EOT | 52.2 | 50.9 | 51.1 | 49.8 | 48.6 | 47.6 |
| CAGE | 49.4 | 46.5 | 45.1 | 43.0 | 43.1 | 42.7 |

*Table 18.* Attack performance (%) using different optimizers on VQA-v2. Lower robust accuracy indicates a stronger attack.

| Method | 576 | 192 | 128 | 64 | 32 | 16 |
|--------|-----|-----|-----|----|----|----|
| VEAttack (APGD) | 56.0 | 55.9 | 55.0 | 54.9 | 51.0 | 49.6 |
| CAGE (APGD) | 51.2 | 48.7 | 46.7 | 45.4 | 43.5 | 42.9 |
| VEAttack (MI-FGSM) | 55.6 | 56.0 | 54.1 | 53.6 | 51.1 | 49.4 |
| CAGE (MI-FGSM) | 48.5 | 46.7 | 46.5 | 44.4 | 42.4 | 43.6 |
| VEAttack (PGD) | 55.8 | 55.7 | 53.8 | 53.5 | 51.3 | 49.7 |
| CAGE (PGD) | 49.4 | 46.5 | 45.1 | 43.0 | 43.1 | 42.7 |

### E.11. Additional Attack Scenario: DriveQA

We further evaluate CAGE on DriveQA to examine its effectiveness in a driving-oriented VQA scenario. This setting is practically relevant because LVLMs are increasingly considered for safety-critical applications such as autonomous driving, where visual token compression may be adopted to reduce inference latency. As shown in Table 19, CAGE consistently achieves lower robust accuracy than VEAttack across all token budgets. For example, under the full-token setting, CAGE reduces robust accuracy from 28.5% to 21.0%. When the model retains only 16 visual tokens, CAGE further reduces robust accuracy from 18.5% to 14.0%.

These results show that the optimization-inference mismatch is not limited to general-purpose VQA benchmarks. Even in a driving-oriented scenario, where the visual questions are more closely related to safety-critical scene understanding, compression-aware perturbation optimization remains more effective than the baseline attack. This further supports the need to evaluate efficient LVLMs with attacks that explicitly account for the compressed visual-token pathway.

## F. Possible Defenses

### F.1. Exisiting Defenses

Existing defenses against visual adversarial attacks on LVLMs can be broadly grouped into two categories. The first category improves robustness by strengthening the vision encoder itself, typically through adversarial training or robust fine-tuning (Schlarmann et al., 2024; Dong et al., 2025). A representative example is RobustCLIP (Schlarmann et al., 2024), which adversarially fine-tunes the CLIP vision encoder to enhance the robustness of downstream systems built upon CLIP features. To examine its effect under visual token compression, we conduct VQA experiments on LLaVA-v1.5-7B while controlling the vision encoder and input resolution. Specifically, we evaluate both the standard and robust victims at $224 \times 224$, keep the multimodal projector and LLM unchanged, and only replace the vision tower with either the standard CLIP ViT-L/14 encoder or the training-based robust CLIP encoder FARE, using consistent preprocessing in both settings. As shown in Table 20, RobustCLIP indeed brings substantial robustness gains across all token budgets. For example, under the full-token setting, the adversarial accuracy increases from 45.1% to 64.9%, while the clean accuracy only slightly decreases from 72.0% to 70.4%. This suggests that robust visual representation learning is an effective way to improve the adversarial robustness of LVLMs. However, dopting RobustCLIP requires replacing and adversarially fine-tuning the visual encoder, which introduces substantial training cost and changes the victim architecture itself.

The second category consists of training-free, inference-time defenses that operate in a black-box manner. A recent representative method is defense through partial-perception supervision (DPS) (Zhou et al., 2025), which uses responses from partial-image perception to supervise the model's answer on the original image. DPS is appealing because it does not require retraining and is applicable to black-box LVLMs. As shown in Table 21, DPS indeed yields non-trivial robustness gains, suggesting that response correction from partial views can mitigate a portion of attacked inputs. However, its inference

*Table 19.* Attack performance (%) on DriveQA under different visual token budgets. Lower robust accuracy indicates a stronger attack.

| Setting | 576 | 192 | 128 | 64 | 32 | 16 |
|---------|------|------|------|------|------|------|
| Clean | 44.5 | 41.5 | 40.0 | 38.0 | 38.5 | 32.0 |
| VEAttack | 28.5 | 24.5 | 22.5 | 18.5 | 18.5 | 18.5 |
| CAGE | 21.0 | 17.5 | 16.0 | 17.5 | 15.5 | 14.0 |

*Table 20.* Defense performance of RobustCLIP.

| Method | 256 (Full) | 128 | 64 | 32 | 16 |
|--------|------------|------|------|------|------|
| Clean Accuracy | 72.0 | 72.2 | 70.4 | 67.5 | 64.5 |
| Adv Accuracy | 45.1 | 44.8 | 43.3 | 42.9 | 41.4 |
| Clean Accuracy w/ RobustCLIP | 70.4 | 69.2 | 69.9 | 66.8 | 64.6 |
| Adv Accuracy w/ RobustCLIP | 64.9 | 64.1 | 62.4 | 59.0 | 58.4 |

cost is substantially higher, as it requires multiple partial-perception queries together with an additional supervision step during decoding. This overhead makes DPS less suitable for the efficiency-sensitive compressed-LVLM setting considered here. Since token compression is introduced precisely to reduce inference cost, a defense that incurs repeated queries and extra generation steps becomes less practical in this regime. Therefore, while DPS is a reasonable generic inference-time defense, its computational overhead weakens its suitability as a defense baseline for compressed LVLMs.

### F.2. Selection-based Selection

Token compression introduces a new attack surface: the model's prediction becomes disproportionately dependent on a small set of surviving tokens. Compression-aligned adversaries can therefore amplify their impact by shaping which tokens pass through the bottleneck and concentrating distortion on these high-impact survivors. This motivates selection-based defenses that intervene at the token selection stage rather than post hoc denoising. In this section, we introduce two *selection-based* defenses that mitigate such predictability from two complementary angles: (i) selecting *robust* tokens whose importance is stable under small perturbations, and (ii) randomizing the survivor set via a *stochastic candidate pool*.

**D1: Robustness-Aware Selection.** Let $a(x) \in \mathbb{R}^{N-1}$ denote the CLS-to-token attention scores at the penultimate attention layer of the vision encoder for an input image $x$, excluding the CLS token, where $N$ is the number of visual tokens including CLS. We estimate attention stability using $M$ noisy views $\{x + \delta_m\}_{m=1}^{M}$, where $\delta_m$ is small i.i.d. Gaussian noise with pixel-level scale $\eta$, and the perturbed image is clipped to the valid pixel range. We define a robustness-aware importance score for token $i \in \{1, \ldots, N-1\}$ as

$$s_i(x) = \mathbb{E}_m[a_i(x + \delta_m)] - \beta \cdot \mathrm{Std}_m[a_i(x + \delta_m)], \tag{13}$$

where $\mathbb{E}_m[\cdot]$ and $\mathrm{Std}_m[\cdot]$ are computed over $m = 1, \ldots, M$, and $\beta$ controls the penalty on attention instability. Tokens are ranked by $s_i(x)$ and the top-$K_{\mathrm{dom}}$ are selected as dominant survivors. In our experiments, we use $M = 4$, $\eta = 1/255$, and $\beta = 2.0$.

**D2: Stochastic Candidate Pool.** To reduce the predictability of deterministic Top-$K$ selection, we introduce a simple randomized bottleneck. Given an importance score $\tilde{s} \in \mathbb{R}^{N-1}$, we first take a slightly larger candidate set by selecting the top-$(K + \Delta)$ tokens, then uniformly sample $K$ survivors from it:

$$\mathcal{C} = \text{Top-}(K + \Delta)(\tilde{s}), \qquad \mathcal{S} \sim \mathrm{Unif}(\mathcal{C}, K),$$

where $\mathrm{Unif}(\mathcal{C}, K)$ denotes sampling $K$ elements from $\mathcal{C}$ without replacement. When $\Delta = 0$, this reduces to deterministic Top-$K$; larger $\Delta$ increases randomness and makes the survivor set harder to predict across runs. Generally, we set $\Delta = K$. In practice, if $K + \Delta \geq N - 1$, the candidate pool covers all non-CLS tokens and the procedure degenerates to uniformly sampling $K$ tokens from the full set (i.e., purely random selection). Formally, we set the effective pool size as $K + \Delta \leftarrow \min(K + \Delta, N - 1)$.

**Experimental Results.** As shown in Table 3, D1 improves robust accuracy under moderate compression budgets but tends to slightly reduce clean accuracy, and it can even hurt robustness in the full-token setting. In contrast, D2 largely preserves

*Table 21.* Defense performance of DPS (Zhou et al., 2025).

| Setting | Full | 192 | 128 | 64 | 32 | 16 |
|---|---|---|---|---|---|---|
| Clean Acc. | 74.5 | 73.4 | 72.1 | 69.3 | 68.4 | 63.2 |
| Robust Acc. | 49.4 | 46.5 | 45.1 | 43.0 | 43.1 | 42.7 |
| Robust Acc. w/ DPS | 64.4 | 64.6 | 63.1 | 61.7 | 57.6 | 49.1 |

*Table 22.* Detection performance using Top-$K$ attention mass as the score under cross-attack evaluation settings (threshold calibrated on VEAttack). We report **Acc.** (Detection accuracy), **TPR** (True Positive Rate, measuring attack detection success), **FPR** (False Positive Rate, measuring false alarms on clean inputs), and the **F1** score.

| $K$ | Acc. | TPR | FPR | F1 |
|---|---|---|---|---|
| 1 | 0.877 | 0.892 | 0.137 | 0.879 |
| 2 | 0.907 | 0.898 | 0.083 | 0.906 |
| 4 | 0.925 | 0.934 | 0.083 | 0.925 |
| 16 | 0.772 | 0.580 | 0.035 | 0.718 |

clean accuracy and provides modest robustness gains at moderate budgets, while becoming ineffective (or even harmful) under the most extreme low-token regime. Overall, although both defenses offer partial improvements, the adversarial robustness remains limited, likely due to an inherent informativeness–robustness trade-off: tokens that are most stable under perturbations are often less semantically informative for downstream tasks.

### F.3. Attention-based Adversarial Detection

In addition to selection-based defenses, we also investigate whether adversarial inputs can be detected from their attention signatures. Our key observation is that adversarial perturbations tend to *reshape* the vision encoder's CLS-to-token attention distribution, making it less concentrated on a small subset of salient tokens. Consequently, the cumulative attention mass carried by the most-attended tokens becomes a discriminative signal for separating clean and adversarial inputs.

Let $a(x) \in \mathbb{R}^{N-1}$ denote the CLS-to-token attention scores. We normalize it into a probability mass function $\hat{a}(x)$ by $\hat{a}_i(x) = a_i(x) / \sum_j a_j(x)$, and then define the *Top-$K$ attention mass* as:

$$m_k(x) = \sum_{i \in \text{Top-}K(\hat{a}(x))} \hat{a}_i(x), \tag{14}$$

where $\text{Top-}K(\hat{a}(x))$ returns the indices of the $K$ largest entries of $\hat{a}(x)$. Intuitively, $m_K(x)$ measures how much attention is concentrated in the most salient $K$ tokens: larger values indicate a more peaked attention pattern.

Following our implementation, we use $\ell(x) = -m_K(x)$ as the adversarial score and classify an input as adversarial if $\ell(x) \geq \tau$. We set $\tau$ using a training split: we randomly sample $40\%$ of the available (clean/adversarial) examples as the training set and choose $\tau$ to maximize the Youden index (TPR$-$FPR) on its ROC curve; the remaining $60\%$ of the data is used for evaluation.

Figure 5 shows that the Top-$K$ mass yields a clear separation between clean and adversarial examples, and the ROC curve indicates strong discriminative power (AUC $\approx 0.966$ on the training split with $K = 4$). Importantly, this detector is lightweight: it only requires reading attention maps from a single forward pass and computing a scalar statistic.

**Experimental Results.** Figure 5 visualizes the distribution of Top-$K$ attention mass and the corresponding ROC curve, showing clear separability between clean and adversarial inputs. On the training split used for threshold selection, the ROC curve yields a strong AUC (e.g., AUC $\approx 0.96$), indicating that attention concentration provides a discriminative signal for detection. Table 4 reports detection performance using the Top-$K$ attention mass as the scoring statistic. Overall, we observe that a *small-to-moderate* $K$ provides the most discriminative signal, while overly large $K$ can degrade robustness by blurring the contrast between clean and adversarial attention patterns. Specifically, $K = 4$ achieves the best overall balance (Acc.=0.940, F1=0.938) with a low false-alarm rate (FPR=0.030), indicating that adversarial examples primarily reduce the concentration of attention on the most salient few tokens. In contrast, increasing $K$ to 16 substantially raises false positives (FPR=0.190) without improving detection sensitivity (TPR remains 0.910), which lowers both Acc. and F1. This suggests

that aggregating attention mass over too many tokens dilutes the "peakiness" cue: clean inputs may still allocate non-trivial attention beyond the top few tokens, making the Top-$K$ mass less separable. Therefore, we use a moderate $K$ (e.g., $K = 4$) as the default setting in our detector.

**Generalization to Unseen Attacks.** To evaluate the practical robustness of the proposed detector, we conduct a cross-attack evaluation where the detection threshold $\tau$ is calibrated using adversarial examples from VEAttack, rather than the target attack itself. As shown in Table 22, this calibration mismatch leads to a notable degradation in performance. Most critically, we observe a sharp increase in the FPR. For instance, at the previously optimal setting of $K = 4$, the FPR rises to $8.38\%$, nearly tripling the rate observed in the self-calibrated setting ($\sim 3\%$). This indicates that the "attention dispersion" boundary learned from VEAttack is ill-suited for the target distribution, causing the detector to aggressively misclassify clean inputs as adversarial. These findings highlight a fundamental limitation: reliance on a single scalar statistic renders the detector brittle to shifts in attack strategies, necessitating the development of more comprehensive, multi-feature detection frameworks.

# G. Limitations and Future Work

While our work provides the first comprehensive study on the adversarial robustness of LVLMs under visual token compression and proposes the effective `CAGE`, several limitations remain that pave the way for future research.

**Scope of Compression Mechanisms.** Our study primarily focuses on *training-free, plug-and-play* visual token compression methods (e.g., VisionZip, VisPruner) applied to off-the-shelf LVLMs. While these represent a widely used paradigm for efficient deployment, we have not explored inner-LLM or training-based compression methods (e.g., FastV (Chen et al., 2024a), Honeybee (Cha et al., 2024) or MobileVLM (Chu et al., 2024)) . Since these methods modify the compression mechanism and the resulting token dynamics in fundamentally different ways, their robustness characteristics may differ from the settings studied here, motivating a systematic extension of our analysis to these alternative compression families.

**Generalization to videos, multi-image inputs, and agentic settings.** We focus on image-based LVLM tasks and a representative set of benchmarks. However, many real-world applications may go beyond single images: models often process videos or multiple images, and are increasingly used in agentic workflows that iterate over observe–reason–act steps. These settings introduce additional challenges, e.g., temporal consistency across frames, cross-image evidence aggregation, and error accumulation over multiple steps, which may interact with token compression in ways not captured by our current evaluation. A natural next step is to study whether compression-aligned attacks remain effective under such inputs and long-horizon decision loops, and to characterize how the compression bottleneck shapes robustness when visual evidence evolves over time.

**More robust defenses.** Our defense study indicates that lightweight detectors based on a single attention statistic and a calibrated threshold can work well when the calibration attack matches the test-time threat model, but their performance drops under attack distribution shifts (e.g., trained on VEAttack, tested on other strategies), revealing limited generalization. This suggests that current defenses for compressed LVLMs are still fragile and can overfit to specific attack signatures. Future work should develop more robust defenses, e.g., combining multi-layer/multi-statistic signals and incorporating the compression bottleneck into robustness objectives.

In summary, we hope this work serves as a stepping stone for designing next-generation LVLMs that are both efficient and secure by design.

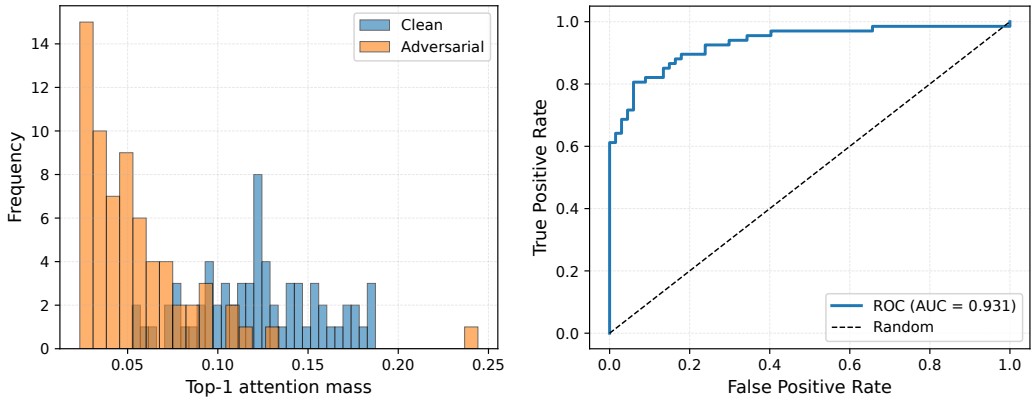

*(a)* Top-1 attention mass distribution (clean vs. adver-*(b)* ROC curve using top-1 attention mass as the detec-
sarial). tion score.

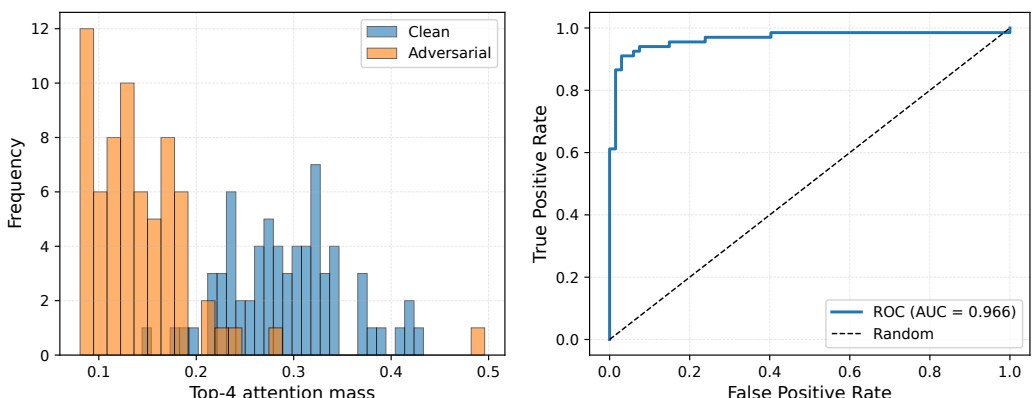

*(c)* Top-4 attention mass distribution (clean vs. adver-*(d)* ROC curve using top-4 attention mass as the detec-
sarial). tion score.

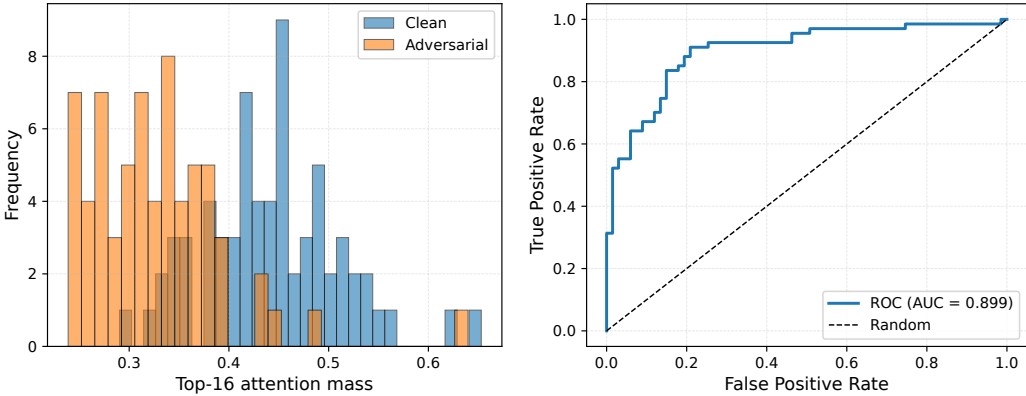

*(e)* Top-16 attention mass distribution (clean vs. adver-*(f)* ROC curve using top-16 attention mass as the detec-
sarial). tion score.

*Figure 5.* Attention-based adversarial detection via the top-k CLS-to-token attention mass.

