# OpenReview forum: "On the Adversarial Robustness of Large Vision-Language Models under Visual Token Compression"
_ICML.cc/2026/Conference — ICML 2026 regular_

### Official Review · Reviewer_6dmZ · 2026-03-04

**Soundness:** 4
**Presentation:** 3
**Significance:** 4
**Originality:** 3
**Overall Recommendation:** 5
**Confidence:** 5

**Summary:**

This paper explores the robustness of LVLMs under visual token compression and proposes a new attack to address the optimization-inference mismatch problem in current attacks. This is the first study to focus on the impact of label compression techniques on robustness and the defensive analysis in this paper provide more directions for future research.

**Compliance With Llm Reviewing Policy:**

Affirmed.

**Final Justification:**

My concerns have been addressed, and I have no further comments.

**Key Questions For Authors:**

(1) Can the authors evaluate whether CAGE generalizes across different vision encoders (e.g., generating adversarial perturbations on encoder A and testing on encoder B under the same compression setting)?

(2) What is the computational overhead of CAGE compared to the baseline？

(3) How does the defense methods compare against a simple random selection baseline?

**Limitations:**

yes

**Strengths And Weaknesses:**

**Strengths**

(1) This research question is crucial and far-reaching as token compression is a key research area in LVLM.

(2) The insights are interesting and reveals the limitations of the current works.

(3) The method is well-motivated and effective across different compression algorithms and datasets.

(4) This paper is clearly structured, fluently written, and easy to understand.

**Weaknesses**

(1) The paper focuses on a grey-box threat model with a known vision encoder, but it lacks a systematic analysis of whether the proposed compression-aligned attack generalizes across different vision encoders.

(2) While the method introduces additional objectives (e.g., EFD and RDA), the paper does not provide a comprasion of the computational overheard.

(3) Table 3 proposes two defenses, where D2 expands a candidate pool and then performs selection. However, it is unclear whether the reported improvements come from the specific selection strategy or simply from increased randomness/diversity. A strong missing baseline is random selection, which could provide a competitive robustness–utility trade-off.

---

> ### Author Rebuttal · Authors · 2026-03-31
>
> Thank you for your time and feedback on our paper.
> >W1+Q1: The paper focuses on a grey-box threat model with a known vision encoder, but it lacks a systematic analysis of whether the proposed compression-aligned attack generalizes across different vision encoders.
>
> **1. Threat model.** Our paper focuses on a gray-box setting, where the attacker knows victim LVLM’s visual encoder. This is a realistic assumption for LVLMs, since many systems are built on public or identifiable vision backbones. Moreover, this threat model allows us to focus on the compression influence, rather than conflating it with other factors such as cross-encoder transferability. Our main claim is therefore made under this threat model.
>
> **2. Fully black-box setting.** Nevertheless, following your suggestions, we evaluate a fully black-box setting using a different surrogate vision encoder. Specifically, attacks are optimized on openai/clip-vit-base-patch16, while evaluation is performed on the target compressed LVLM under VisionZip with different token budgets. For both the baseline and CAGE, we use the same attack setting: $\epsilon=16/255$, step size $\alpha=1/255$, and 100 PGD steps.
> The fully black-box setting is substantially more challenging than the gray-box setting, and the overall attack performance becomes much lower for both methods. Under the fully black-box setting, the additional gain of CAGE over the baseline also becomes smaller, but CAGE still achieves lower robust accuracy under most compressed budgets. This is expected because the attacker must transfer not only the perturbation itself across encoders, but also the compression-aware signal, i.e., its alignment with the token-selection and retention behavior of the victim compressed LVLM. The reduced gain reflects the shared difficulty of cross-encoder transfer in existing black-box attacks, whereas our paper focuses on a different question: how token compression changes adversarial robustness and introduces an optimization–inference mismatch under the gray-box setting. Extending gray-box attacks (including ours) to the fully black-box setting remains an important and challenging direction.
>
> Table1: Attack performance under a black-box setting
> | Method | 576 (Full) | 192 | 128 | 64 | 32 | 16 |
> |---|---:|---:|---:|---:|---:|---:|
> | Clean | 74.5 | 73.4 | 72.1 | 69.3 | 68.4 | 63.2 |
> | Baseline | 69.5 | 69.4| 68.7 | 65.5 | 64.4  | 63.0 |
> |CAGE | 69.4 | 69.1 | 66.9 | 64.8 | 63.1 | 62.2  |
>
> >W2+Q2: The paper does not provide a comprasion of the computational overheard.
>
> We report the runtime of different attacks on a single NVIDIA RTX 4090. As shown below, CAGE introduces only marginal additional overhead compared with VEAttack, while remaining substantially more efficient than VT-Attack. These results suggest that the stronger attack performance of CAGE does not come at a high runtime cost.
>
> Table2: Runtime comparison of CAGE with current attacks
> | Method       | Avg. time / image (s) | Relative cost vs. VEAttack |
> |--------------|----------------------:|---------------------------:|
> | VEAttack     | 5.35                  | 1.00×                      |
> | AttackVLM-ii | 5.28                  | 0.99×                      |
> | VT-Attack    | 9.63                  | 1.80×                      |
> | CAGE         | 5.74                  | 1.07×                      |
>
> >W3+Q3:  A strong missing baseline is random selection, which could provide a competitive robustness–utility trade-off.
>
> We additionally include a random selection baseline to examine whether the gain mainly comes from increased randomness/diversity. The results are shown below. We observe that random selection does provide some robustness gain, suggesting that additional diversity/randomness is indeed helpful. At the same time, D2 generally achieves higher clean accuracy while maintaining comparable robust accuracy relative to this random baseline. We will include this baseline and clarify this point in the revision.
>
> Table3. Defense performance of random selection
> | Setting | Full | 192 | 128 | 64 | 32 | 16 |
> |---|---:|---:|---:|---:|---:|---:|
> | Clean Acc. | 74.5 | 73.4 | 72.1 | 69.3 | 68.4 | 63.2 |
> | Robust Acc. | 49.4 | 46.5 | 45.1 | 43.0 | 43.1 | 42.7 |
> | Clean Acc. (Random) | 74.4 | 71.6 | 71.2 | 67.9 | 66.9 | 61.5 |
> | Robust Acc. (Random) | 50.6 | 48.4 | 47.7 | 46.5 | 44.9 | 42.8 |

---

> > ### Author Rebuttal · Reviewer_6dmZ · 2026-04-01
> >
> > My concerns have been addressed, and I have no further comments.

---

> > > ### Author Response · Authors · 2026-04-01
> > >
> > > Dear reviewer, thank you so much for your positive feedback and for raising the score to acceptance! It encourages us a lot! We will ensure that the revised version includes the new experiments and corresponding discussions.

---

### Official Review · Reviewer_6Ani · 2026-03-04

**Soundness:** 3
**Presentation:** 4
**Significance:** 3
**Originality:** 3
**Overall Recommendation:** 5
**Confidence:** 4

**Summary:**

This paper addresses adversarial attacks against LVLM inference pipelines that incorporate a visual token pruning mechanism. It is shown that existing attacks unaware of the compression are still effective, however may overestimate robustness. A specialized attack ("CAGE") is proposed and it is shown that it consistently outperforms the baseline attack. The proposed attack incorporates two losses: the EDF loss focusses on applying perturbations to tokens that likely survive the pruning, while the RDA loss focusses on increasing the selection probability of highly distorted tokens.

**Compliance With Llm Reviewing Policy:**

Affirmed.

**Final Justification:**

The rebuttal addressed my concerns and questions.

**Key Questions For Authors:**

1. In the RDA loss (equation (11)), why are gradients through p^s stopped? Isn't the point of the RDA loss to increase selection scores for highly distorted tokens?
2. Which exact LLaVA model is used in the experiments? (e.g. LLaVA 1.5 7B?)

**Limitations:**

yes

**Strengths And Weaknesses:**

**Strengths**

1. The manuscript is well written and easy to follow.
2. Comprehensive experiments with various compression methods are conducted. The proposed CAGE attack outperforms the baseline across settings.
3. The method and its parts are adequately ablated, in particular the two loss components and their weights.

**Weaknesses**

1. Experiments focus on the LLaVA model, which is not very recent. In the appendix are also results for Qwen 2.5 VL, however this is also not the newest generation of models. An analysis on more recent models would be interesting.
2. From my understanding, attacks are always conducted with 100 iterations. It would be interesting to see whether CAGE maintains its advantage over the baseline when using more iterations.
3. A discussion on the runtime of CAGE vs. the baseline would be interesting.

**Minor**

1. Algorithm 1
	- Typo: L_RDS instead of L_RDA
	- It is not noted that the gradient through one component of L_RDA is stopped
2. Selection and detection based defenses against the proposed attack are discussed in Section 6, however training based defenses  should also be mentioned, e.g. [A].

[A] Robust CLIP: Unsupervised Adversarial Fine-Tuning of Vision Embeddings for Robust Large Vision-Language Models, ICML 2024

---

> ### Author Rebuttal · Authors · 2026-03-31
>
> Thank you for your time and feedback on our paper.
> >W1: An analysis on more recent models would be interesting.
>
> We mainly use LLaVA for two reasons. First, most existing visual token compression methods **only release implementations on LLaVA** since its modular vision-language pipeline makes compression easy to implement and compare. Second, many newer LVLMs introduce additional built-in visual-token compression mechanisms, making it harder to isolate the effect of the external compression pipeline studied in our paper.
>
> To further broaden the evaluation, we extend the experiments to InternVL3-8B with different compression methods, and we report the corresponding VQA-v2 results as follows, which give the same conclusion as LLaVA.
>
> Table1: The experimental results on InternVL3
> | Compression | k | Clean | Baseline | CAGE |
> |-------------|--:|---:|---:|---:|
> | -   | 144 | 71.1 | 43.6 | 40.0 |
> | VisionZip   |  72 | 70.8 | 39.9 | 33.4 |
> | VisionZip   |  36 | 66.0 | 35.5 | 30.8 |
> | VisPruner   |  72 | 69.7 | 40.8 | 35.9|
> | VisPruner   |  36 | 66.0 | 36.7 | 33.3 |
> | DivPrune    |  72 | 69.5 | 41.6 | 38.1 |
> | DivPrune    |  36 | 66.3 | 38.3 | 34.4 |
>
> >W2: Whether CAGE maintains its advantage over the baseline when using more iterations.
>
> We further increase PGD iterations from 100 to 200 on VQA-v2. As shown below, CAGE still achieves lower robust accuracy than the baseline across all token budgets.
>
> Table2: Attack performance under 200 steps
> | Method   | 576 | 192  | 128  | 64   | 32   | 16   |
> |----------|-----------:|-----:|-----:|-----:|-----:|-----:|
> | VEAttack |       51.7 | 50.1 | 50.4 | 49.2 | 47.8 | 46.7 |
> | CAGE     |       47.3 | 45.0 | 44.0 | 43.4 | 43.8 | 42.0 |
>
> >W3: A discussion on the runtime of CAGE vs. the baseline would be interesting.
>
> We compare the runtime of different attacks on a single NVIDIA RTX 4090. As shown below, CAGE introduces only marginal additional overhead compared with VEAttack, while remaining substantially more efficient than VT-Attack.
>
> Table3: Runtime comparison of CAGE with current attacks
> | Method       | Avg. time / image (s) | Relative cost vs. VEAttack |
> |-|-|-|
> | VEAttack     | 5.35                  | 1.00×                      |
> | AttackVLM-ii | 5.28                  | 0.99×                      |
> | VT-Attack    | 9.63                  | 1.80×                      |
> | CAGE         | 5.74                  | 1.07×                      |
>
>
> > Minor1: Algorithm 1
>
> We will fix the typo to L_RDA and explicitly annotate in Algorithm 1 that gradients through the selection-score branch are stopped.
>
> > Minor2: Training-based defenses should also be mentioned.
>
> We additionally discuss existing defenses from the following two perspectives.
> 1. **Training-based defenses** that improve the vision encoder itself (e.g., RobustCLIP). We note that RobustCLIP is not directly applicable as a clean defense in our gray-box setting. It improves robustness by adversarially fine-tuning the vision encoder, which changes not only the victim model’s visual backbone but also the surrogate model used for attack generation. As noted in prior work (such as VEAttack), using a more robust surrogate does not necessarily weaken transfer-based attacks, and may even increase attack effectiveness.
> 2. **Training-free inference-time defenses** such as DPS[1]. Following the reviewer WGEY’s suggestion, we additionally evaluate DPS, and the results are shown below. DPS indeed provides non-trivial robustness gains. However, it also introduces substantial inference overhead, as it requires multiple partial-perception queries and an additional supervision step during decoding. Since our setting focuses on compressed LVLMs motivated by efficiency, this extra cost limits the practicality of DPS.
>
> Table4. Defense performance of DPS
> | Setting | Full | 192 | 128 | 64 | 32 | 16 |
> |-|-|-|-|-|-|-|
> | Robust Acc. | 49.4 | 46.5 | 45.1 | 43.0 | 43.1 | 42.7 |
> | Robust Acc. w/ DPS | 64.4 | 64.6 | 63.1 | 61.7 | 57.6 | 49.1 |
>
> [1]Defending LVLMs Against Vision Attacks through Partial-Perception Supervision, ICML25.
>
> >Q1: In the RDA loss (equation (11)), why are gradients through $p^s$ stopped?
>
> We apologize for the confusion. This is a typo in the paper: the detached term should be $p^d$, not $p^s$. We will correct Eq. (11) and the corresponding description in the revision.
>
> >Q2: Which exact LLaVA model is used in the experiments? (e.g. LLaVA 1.5 7B?)
>
> The main LLaVA experiments are conducted on liuhaotian/llava-v1.5-7b. We will specify this explicitly in the paper.

---

> > ### Author Rebuttal · Reviewer_6Ani · 2026-04-02
> >
> > I'd like to thank the authors for the rebuttal, which addresses my concerns. Thereby, I will raise my score. However, I do not follow their argument that a training based defense is not applicable or not effective. It is my understanding that the cited VEAttack paper shows that robust victim models are more robust against transfer based attacks than clean models. I agree that using a robust surrogate model will generally increase effectiveness when transferring to a robust victim model, however robust victim models still maintain higher performance than clean victim models, irrespective of the employed surrogate model. Thus, training based defenses would be beneficial for model providers in order to defend against the proposed attack.

---

> > > ### Author Response · Authors · 2026-04-03
> > >
> > > Dear reviewer, thank you very much for your positive feedback and for raising the score to acceptance. We are truly encouraged by your support, and we will make sure that the revised version includes the new experiments and corresponding discussion.
> > >
> > > We agree with your point that robust victim models can still maintain higher robustness than clean victim models, regardless of the surrogate model used for transfer-based attacks. To further clarify this point, we added additional experiments to examine the effect of training-based defenses in our setting. Concretely, we conduct VQA experiments on LLaVA-v1.5-7B while controlling the vision encoder and input resolution. We evaluate both the clean and robust victims at 224×224, keep the multimodal projector and LLM unchanged, and only replace the vision tower, using either standard CLIP ViT-L/14 or the training-based robust CLIP model FARE, with consistent preprocessing in both settings.
> > > As shown in the table below, RobustCLIP is indeed an important way to improve robustness, although it requires substantial training cost.
> > >
> > >
> > > Table 1. Defense performance of RobustCLIP
> > > | Method | 256 (Full) | 128 | 64 | 32 | 16 |
> > > |---|---:|---:|---:|---:|---:|
> > > | Clean accuracy | 72.0 | 72.2 | 70.4 | 67.5 | 64.5 |
> > > | Adv accuracy | 45.1 | 44.8 | 43.3 | 42.9 | 41.4 |
> > > | Clean accuracy under RobustCLIP | 70.4 | 69.2 | 69.9 | 66.8 | 64.6 |
> > > | Adv accuracy under RobustCLIP | 64.9 | 64.1 | 62.4 | 59.0 | 58.4 |
> > >
> > >
> > > We will include these results and the corresponding discussion in the revised paper. Thank you again for this valuable point!

---

### Official Review · Reviewer_WGEY · 2026-03-14

**Soundness:** 2
**Presentation:** 2
**Significance:** 2
**Originality:** 2
**Overall Recommendation:** 4
**Confidence:** 4

**Summary:**

The paper studies test-time attacks against large vision language models (LVLMs). It finds that prior attacks often fail to mislead LVLMs when token-compression methods are employed. To address this, the paper proposes CAGE, an attack that focuses the adversarial perturbations on tokens that are most likely to survive compression, tailored for settings where the vision encoder is known to the adversary, but the number of tokens surviving compression (K) and to the language model are unknown. Experiments with five recent compression methods, three datasets, and two models demonstrate superior attack success compared to a recent attack.

**Compliance With Llm Reviewing Policy:**

Affirmed.

**Final Justification:**

The authors did an excellent job addressing my concerns in the rebuttal. Accordingly, I decided to increase my recommendation from “reject” to “weak accept.”

**Key Questions For Authors:**

- Why do untargeted attacks produced by CAGE pose a realistic security risk?

- Could you justify the use of PGD and the choice of parameters?

**Limitations:**

As discussed in the weaknesses section above, the limitations of the work do not seem to be adequately discussed.

**Strengths And Weaknesses:**

Thank you for submitting your paper to ICML 2026. I found it enjoyable to read. However, the paper has several critical weaknesses that prevent me from recommending its acceptance.

## Strengths

+ **Weak threat model, with little assumptions.** The attack only requires access to the vision encoder.

+ **Experiments with diverse compression methods.** The paper evaluates the proposed methods against a diverse set of recent compression methods published in top venues.

+ **CAGE outperforms VEAttack.** The proposed attack attains substantially higher success rates than the recent Mei et al. attack (ICLR 2026).

## Weaknesses

- **Inadequate metrics for assessing success.** Checking for whole word containment seems inadequate. Shouldn’t an attack attempt that produces an output with no exact word match with the ground truth but where the semantic meaning of the output is equivalent to the ground truth be considered a failed attempt?

- **Only one baseline attack.** Although the space is heavily studied, only one attack against LVLMs (Mei et al.’s) is considered as a baseline.

- **Focus on perturbations with bounded $L_\infty$ norm.** Since $L_\infty$-bounded perturbations have equal budgets across all tokens, I’m unconvinced that there’s a need to “focus the perturbations on the tokens most likely to survive compression.” Such an objective seems more appropriate for perturbations bounded in $L_2$ or $L_1$ norms. I also wonder if an attack that follows an expectation-over-transformation-like approach would be more adequate for $L_\infty$.

- **Unjustified choice of attacks and parameters.** The paper doesn’t justify the use of a relatively dated attack (PGD) with a specific set of hyperparameters (e.g., $\epsilon$=2/255).

- **Limited threat model with unclear security implications.** The paper only considers untargeted attacks and doesn’t clarify the real-world security risk of the attacks it studies. It should make the implications clearer and consider targeted attacks, which are likely to pose a higher security risk.

- **No evaluation of existing defenses.** The paper doesn’t evaluate CAGE against existing defenses (e.g., https://arxiv.org/abs/2412.12722 from ICML 2025).

- **No universal vision encoder used.** The experiments would be more compelling if a universal vision encoder (per the threat model) is used across the different LVLMs.

- **The presentation is lacking.** There’s significant room for improving the presentation. Some examples: (1) EFD and RDA (the primary methodological components of CAGE) are not described clearly in the introduction; (2) the constraints on $\delta$ aren’t properly discussed in Section 2.2; (3) in Section 2.1, I’m missing a background on different compression approaches; (4) the analysis in Section 3 shows how compressors affect attacks (e.g., by selecting the most heavily selected tokens), but doesn’t explain why; (5) there’s a significant lie factor in the graphs of Figure 4.

---

> ### Author Rebuttal · Authors · 2026-03-31
>
> Thank you for your time and feedback on our paper.
> >W1: Inadequate metrics.
>
> We agree that exact-match metrics might miss some semantically equivalent cases. However, keyword-/answer-matching **remains the standard in current VQA robustness evaluation [1,2]**. This issue is mitigated in our benchmarks because TextVQA typically requires exact text recognition, while VQA-v2 and GQA provide multiple ground-truth answers. Since our goal is to compare attacks, a uniform exact-matching protocol provides a fair and controlled evaluation.
> >W2: Only one baseline.
>
> Beyond VEAttack, we additionally include two more attacks: AttackVLM-ii and VT-Attack, and CAGE still achieves lower robust accuracy than all baselines. We will incorporate the results into the revision.
>
> Table1. Comparison with more baselines
> | Method | 576 | 192 | 128 | 64 | 32 | 16 |
> |-|-|-|-|-|-|-|
> | AttackVLM-ii | 62.6 | 62.1 | 62.1 | 60.8 | 57.3 | 52.5 |
> | VT-Attack | 59.4 | 60.2 | 60.1 | 59.2 | 54.7 | 51.1 |
> | CAGE | 49.4 | 46.5 | 45.1 | 43.0 | 43.1 | 42.7 |
>
> >W3: Perturbations with bounded $L_\infty$ norm.
>
> 1. Focusing perturbations is still meaningful under $L_\infty$, which only constrains the maximum perturbation per pixel, while the actual perturbation remains non-uniform and gradient-driven. We adopt $L_\infty$ because it is **the standard threat model in existing LVLM attacks [1,3,4]**. Using $L_1$ or $L_2$ would introduce a different perturbation geometry and threat model, making results not directly comparable.
> 2. EOT is not specific to the $L_\infty$ setting but a general way to handle uncertainty. In our case, the uncertainty comes from token compression rather than the norm itself. Directly applying EOT would treat compression as a stochastic process by sampling token subsets or budgets and averaging the loss. However, token selection is not arbitrary, but structured and coupled with the model’s attention and encoder. Ignoring this structure leads to inefficient and noisy optimization. Our method instead models token survival probabilities explicitly, capturing this structure in a more efficient way.
>
> > W4+Q2. Choice of attacks and parameters.
>
> 1. PGD is **a standard and widely used optimization method in adversarial robustness**. Existing LVLM attacks also use PGD as the backbone [1,3], making it a natural choice. Using a different optimizer would make comparisons less direct.
> 2. The hyperparameters (e.g., $\epsilon$) follow common practice [1]. They are shared attack settings for all compared methods. Using the same optimizer and perturbation budget keeps all attacks under the same noise constraint, enabling a fair and directly comparable evaluation. We also verify our results under different $\epsilon$ values in the appendix.
>
> > W5+Q1: Unclear security implications.
>
> Untargeted attacks already pose realistic risks. In many LVLM applications, an attacker does not need to force a specific output. Causing incorrect or misleading responses is already harmful. For example, in autonomous driving scenarios, the attacked model may misjudge which lane a vehicle is in, miscount nearby cyclists, or produce inaccurate understanding of road-crossing cues [4,5]. These errors are sufficient to mislead downstream perception and decision-making.
> >W6: Evaluation of existing defenses.
>
> We evaluate DPS, which improves robustness but incurs substantial inference overhead due to multiple queries. This limits its practicality in our efficiency-focused setting.
>
> Table4. Defense performance of DPS
> | - | Full | 192 | 128 | 64 | 32 | 16 |
> |-|-|-|-|-|-|-|
> | Robust Acc. | 49.4 | 46.5 | 45.1 | 43.0 | 43.1 | 42.7 |
> | Robust Acc. w/ DPS | 64.4 | 64.6 | 63.1 | 61.7 | 57.6 | 49.1 |
>
> >W7: No universal vision encoder used.
>
> We emphasize that our paper focuses on a gray-box setting, which is a practical assumption in prior LVLM attacks[1,3], since many LVLMs adopt public or well-known vision encoders. This also allows us to focus on the effect of compression without conflating it with cross-encoder transferability. Refer to W1 of Reviewer 6dmZ for additional black-box results.
>
> >W8: The presentation is lacking.
>
> Thanks a lot for the suggestions. We revise the paper to improve clarity by better introducing EFD/RDA, clarifying the attack constraints, adding a brief background on compression methods, and strengthening the intuition. We also update Fig.4 to better reflect the underlying trends, following Reviewer FEB3’s suggestion.
>
> [1]VEAttack: Downstream-agnostic Vision Encoder Attack against Large Vision Language Models, ICLR26.
>
> [2]InstructBLIP: Towards General-purpose Vision-Language Models with Instruction Tuning, NeurIPS23.
>
> [3]Break the Visual Perception: Adversarial Attacks Targeting Encoded Visual Tokens of Large Vision-Language Models, MM24.
>
> [4]Towards Trustworthy Autonomous Vehicles with Vision-Language Models Under Targeted and Untargeted Adversarial Attacks, CVPR25.
>
> [5]Black-Box Adversarial Attack on Vision Language Models for Autonomous Driving, Arxiv25.

---

> > ### Author Rebuttal · Reviewer_WGEY · 2026-04-01
> >
> > Thank you for the rebuttal! The response partially addressed some of my concerns, but key ones remain unresolved.
> >
> > > W3
> >
> > I’m not convinced that perturbing certain pixels by amounts $<\epsilon{}$ could be beneficial for attacks under the $L_\infty$ constraints. Regardless, experimenting with other types of constraints (such as ones in $L_2$-norm) would better match the assumptions in the paper and further strenghten the work. Moreover, as the different compression methods impose a certain distribution over the VLM inputs for a given image, I disagree with the claim that EOT is inadequate as an attack method.
> >
> > > W4
> >
> > The fact that prior work used a sub-optimal attack isn’t a good reason for adopting that attack. The findings may be differ dramatically if a stronger attack is used.
> >
> > > W5
> >
> > Autonomous driving is a good motivation for untargeted attacks—it’d have been useful to experiment with this use case rather than only suggesting it.

---

> > > ### Author Response · Authors · 2026-04-02
> > >
> > > Thanks for your further comments. We sincerely hope the following clarifications can address your concerns.
> > >
> > > >W3
> > >
> > > **$L_2$.** Following your suggestion, we additionally evaluate CAGE under an $L_2$ threat model. As shown in Table 1, $L_2$ does not provide a larger advantage for CAGE over VEAttack. In fact, the gain is limited and clearly smaller than under $L_\infty$. This is because LVLM behavior is driven by token-level semantic representations rather than individual pixels. While $L_2$ may concentrate more energy on a few sensitive pixels, this does not necessarily translate into a stable semantic shift of the corresponding token. By contrast, under $L_\infty$, the attack more naturally relies on the overall perturbation pattern across a token region, which can induce a more spatially redundant token-level distortion that better survives tokenization and compression. Therefore, in compressed LVLMs, what matters is not simply stronger pixel-level concentration, but whether the perturbation pattern can reliably alter the representations of important tokens.
> > >
> > > To better understand this difference, we further analyze the $L_\infty$ case. Although $L_\infty$ bounds each pixel perturbation, token-level feature change still depends on how perturbations are distributed within a token region. In CLIP patch feature space, CAGE concentrates feature distortion more strongly: the top-5% most affected patch tokens account for 37.4% of the total feature-distortion energy, compared with 29.4% for VEAttack. At the same time, the pixels within these top-5% patches account for only about 4.3% of the total pixel-level perturbation energy. This suggests that **the gain does not come from larger pixel perturbation magnitudes, but from how perturbations are organized within token regions.**
> > >
> > > Table 1. Attack performance under $L_2 < 3$
> > > | Method | 576 | 192 | 128 | 64 | 32 | 16 |
> > > |---|---:|---:|---:|---:|---:|---:|
> > > | VEAttack | 69.5 | 68.0 | 66.0 | 64.0 | 60.2 | 58.3 |
> > > | CAGE | 68.7 | 66.0 | 64.0 | 61.0 | 58.6 | 56.6 |
> > >
> > > **EOT.** We agree that EOT is a natural way to address unknown deployed budgets, and we implement such a baseline by sampling multiple candidate top-K budgets at each iteration and optimizing the average loss over the corresponding retained tokens. As shown in Table 2, CAGE consistently outperforms this baseline across all evaluated token budgets on VQA-v2. The **key difference** is that EOT only treats $K$ as a random variable and optimizes a Monte Carlo approximation of the expected loss, whereas CAGE explicitly models token survival probabilities under budget uncertainty. This yields a more structured and stable compression-aware objective.
> > >
> > > Table 2. Attack performance on VQA-v2
> > > | Method | 576 | 192 | 128 | 64 | 32 | 16 |
> > > |-|-|-|-|-|-|-|
> > > | EOT | 52.2 | 50.9 | 51.1 | 49.8 | 48.6 | 47.6 |
> > > | CAGE | 49.4 | 46.5 | 45.1 | 43.0 | 43.1 | 42.7 |
> > >
> > > > W4
> > >
> > > Following your suggestion, we additionally evaluate stronger optimizers, including APGD and MI-FGSM. The results are shown below. We find that **CAGE consistently outperforms VEAttack under these optimizers as well**, while newer optimizers such as APGD do not necessarily improve attack performance in our setting. Therefore, although PGD may appear classical, it remains a standard and strong optimizer in adversarial robustness evaluation. Many later improvements are mainly developed for **standard classification objectives**, whereas our setting uses a **feature-level objective** for LVLMs. As a result, these optimizers are not necessarily better when transferred directly to our setting. Besides, **our focus is the attack objective rather than a particular optimizer**: PGD is used only as a standard optimization method, while the key comparison in our study is whether explicitly accounting for compression in the optimization objective improves attack effectiveness.
> > >
> > > Table 3. Attack performance using more optimizers
> > > | Method | 576 | 192 | 128 | 64 | 32 | 16 |
> > > |-|-|-|-|-|-|-|
> > > | VEAttack (APGD) | 56.0 | 55.9 | 55.0 | 54.9 | 51.0 | 49.6 |
> > > | CAGE (APGD) | 51.2 | 48.7 | 46.7 | 45.4 | 43.5 | 42.9 |
> > > | VEAttack (MI-FGSM) | 55.6 | 56.0 | 54.1 | 53.6 | 51.1 | 49.4 |
> > > | CAGE (MI-FGSM) | 48.5 | 46.7 | 46.5 | 44.4 | 42.4 | 43.6 |
> > > | VEAttack (PGD) | 55.8 | 55.7 | 53.8 | 53.5 | 51.3 | 49.7 |
> > > | CAGE(PGD) | 49.4 | 46.5 | 45.1 | 43.0 | 43.1 | 42.7 |
> > >
> > >
> > > >W5
> > >
> > > Following your suggestion, we conduct experiments on DriveQA, a driving-oriented VQA benchmark. As shown in Tab. 4, our untargeted attack remains highly effective, and CAGE consistently outperforms VEAttack across all token budgets. This further supports the practical risk of untargeted attacks in safety-critical settings.
> > >
> > > Table 4. Attack performance on DriveQA
> > > | Setting | 576 | 192 | 128 | 64 | 32 | 16 |
> > > |-|-|-|-|-|-|-|
> > > | Clean | 44.5 | 41.5 | 40.0 | 38.0 | 38.5 | 32.0 |
> > > | VEAttack | 28.5 | 24.5 | 22.5 | 18.5 | 18.5 | 18.5 |
> > > | CAGE| 21.0 | 17.5 | 16.0 | 17.5 | 15.5 | 14.0 |

---

### Official Review · Reviewer_FEB3 · 2026-03-16

**Soundness:** 2
**Presentation:** 3
**Significance:** 2
**Originality:** 2
**Overall Recommendation:** 4
**Confidence:** 5

**Summary:**

This paper explores the adversarial vulnerability of LLaVA under the visual token compression. The authors further propose CAGE which consists of EFD and RDA as a new adversarial attack against visual token compression and exalute it on VQA datasets.

**Compliance With Llm Reviewing Policy:**

Affirmed.

**Final Justification:**

I appreciate the effort of authors in addressing all my concerns. I have adjusted my scores accordingly.

**Key Questions For Authors:**

The R/C curve is confusingly difficult to understand. First of all, the use of R/C curve is not straight forward compared to directly using the performance drop, i.e. 1 - R/C, which directly indicates how attacks behave under different K. And the x-axis will be clearer if it follows a decreasing order from full dimension (576) to 16, making the graph more intuitive. In the paper, the author states that as I quote  *As the deployment budget decreases from mild to moderate compression (e.g.,192/128/64tokens), the advantage typically increases*, which is not the case from my understanding that the two curves are bascally in parallel as K decreases, indicating that both CAGE and the baseline become less effective under limited budget. I hope the author could clarify on this confusion and contradictory conclusion.

**Limitations:**

As discussed in weaknesses, the paper lacks certain significance and solidity: the gap between mismatch token number is **not significant enough as a critical challenge**, and the effectivness of CAGE is **not significant enough to be considered fully addressing this issue**.

**Strengths And Weaknesses:**

**Strength**: The paper offers an interesting and valuable perspective of adversarial robustness, bringing insights to how adversarial attacks interact under visual token compression.

**Weakness**: While the overall idea and perspective is interesting, the paper lacks overall depths and solidity regarding its academic contribution.

a.**Motivation**: The author states that existing attacks *overestimate the robustness of compressed LVLMs*. While I do understand that there are few works working on attacking compressed VLMs, I did not see any attack claiming to be identically effective for compressed VLMs. As such, would the authors provide references for such statement? Furthmore, the quoted statement is quite confusing as it conveys the idea that compressed VLMs is not as robust as attackers think, meaning that compressed VLMs are quite easier to attack. However, the paper seems to regard such an attacking scenario as a challenge, which is a little bit contradictory. If I am understanding this paper correctly, should the statement be **the attackers underesitmate the robustness of compressed LVLMs** ?

b. **Effectiveness**: Another issue that undermines the solidity of this paper is its effectiveness. The results in Table.1, which is the motivation for the paper, showing the influence of mismatch, is not significant enough to be regarded as unacceptable performance decreases from attackers perspective. In fact, results for $K=192/64$ show no outstanding difference caused by the mismatch as the two settings have very similar performance. As for the main results on LLaVA and on Qwen (in the supp.), the performance brought by CAGE is also not significant enough, especially on Qwen.

c. **Depths\&Boradness**. While the paper offer certain intuitive insights on attacking compressed LVLMs, the conclusions are primarily drawn from 1 model (LLaVA) and 1 task (VQA), which making its broader applicability weak. It would be recommended to includer a few other LVLMs to fortify such a conlusion. As for tasks, the author uses VQA as the only included task with no evident explaination, while the orignal work of VEAttack covers several tasks in cluding image captioning, VQA and retrieval, not to mention models.

---

> ### Author Rebuttal · Authors · 2026-03-31
>
> Thank you for your time and feedback on our paper.
> >W1: Motivation.
>
> Our intended claim is that existing attacks overestimate the robustness of compressed LVLMs because they are suboptimal for this setting, rather than because prior work explicitly claimed identical effectiveness on compressed models.
>
> 1. Here, “overestimate” refers to the evaluation outcome, not an intrinsic property of the model. We **do not claim** that compressed LVLMs are inherently easier to attack. Rather, if an attack is suboptimal, it yields weaker attack performance and thus higher robust accuracy, making the compressed LVLM appear more robust than it actually is.
> 2. This statement is supported by our experimental results. Under the same compressed setting, VEAttack yields higher robust accuracy than CAGE, indicating that it does not fully expose the vulnerability of compressed LVLMs. Therefore, evaluations based only on such non-compression-aware attacks can be overly optimistic.
>
> To avoid misunderstanding, we will revise the wording to: existing attacks, without considering the compression mechanisms, cannot fully disclose the robustness vulnerabilities of compressed LVLMs.
>
> >W2: Effectiveness.
>
> 1. **The mismatch effect in Table 1 is non-trivial.** It should not be judged only from the $K_{\text{model}}=192$ and $64$ cases. The effect becomes clearer under stronger compression, where mismatch matters more. For example, when $K_{\text{model}}=64$, using $K_{\text{attack}}=16$ gives a robust accuracy of 56, whereas the aligned setting $K_{\text{attack}}=64$ reduces it to 48.7, a gap of 7.3 points. This is already substantial from an attacker’s perspective and highlights the practical challenge that the deployment token budget is typically unknown in advance.
> 2. **CAGE achieves meaningful and consistent gains.** On LLaVA, CAGE consistently reduces robust accuracy across all token budgets and compression methods. For example, on TextVQA, the average robust accuracy drops from 33.5 to 23.4 at $K_{\text{model}}=192$ and from 25.5 to 15.7 at $K_{\text{model}}=16$. We believe these reductions are already non-trivial from an attacker’s perspective. For Qwen2.5-VL, the gain is moderate because its native token merging already partially aligns the baseline attack with the compressed inference pathway, reducing the mismatch that motivates CAGE. Even so, CAGE still consistently achieves lower robust accuracy than the baseline across datasets and token budgets. This result further supports our main point: attack–compression mismatch is an important reason why existing attacks are suboptimal on compressed LVLMs.
>
> >W3: Depths&Boradness.
>
> 1. **Scope compared with VEAttack**. VEAttack studies **generic LVLM** robustness, whereas our paper focuses specifically on robustness under **compressed LVLM** inference. Accordingly, we evaluate the models and tasks that are **standard in the visual token compression literature**. Current public compression baselines and implementations are still mainly built on LLaVA, since its modular vision-language pipeline makes compression easy to implement, and most prior methods are only publicly released on LLaVA. Existing compression work is also predominantly evaluated on VQA tasks, since they are highly sensitive to visual evidence loss and thus provide a suitable testbed for studying the impact of token compression.
> 2. **Broader evaluation beyond the main setting**. Nevertheless, following your suggestions, we extend beyond LLaVA/Qwen+VQA in two directions: (i) another LVLM (InternVL3-8B), where CAGE consistently achieves lower robust accuracy across compression methods (**Refer to response to W1 of Reviewer 6Ani**); (ii) another task (image captioning on COCO), where CAGE consistently obtains lower CIDEr across all token budgets.
>
> Table1. Attack performance on image captioning
> | Method | 576 | 192 | 128 | 64 | 32 | 16 |
> |-|-|-|-|-|-|-|
> | Clean | 1.28 | 1.29 | 1.30 | 1.18 | 1.01 | 0.69 |
> | Baseline | 0.22 | 0.23 | 0.23 | 0.20 | 0.17 | 0.12 |
> | CAGE | 0.17 | 0.17 | 0.13 | 0.11 | 0.12 | 0.09 |
>
> >Q: The R/C curve is confusingly difficult to understand.
>
> 1. We will plot $1 - R/C$ and reorder the x-axis from 576 to 16 to improve clarity.
> 2. We clarify that the “advantage” refers to the relative reduction w.r.t. the baseline, i.e., $(R_{\text{base}} - R_{\text{CAGE}})/R_{\text{base}}$. In contrast, Figure 4 plots $R/C$, where the gap corresponds to $(R_{\text{base}} - R_{\text{CAGE}})/C$. Since $C$ is larger and varies with the token budget, this normalization compresses the gap and makes the improvement less visually pronounced. We also note that the curves are not completely parallel. In particular, on TextVQA, the gap between CAGE and the baseline becomes more pronounced from 576 to 64 tokens. Therefore, the relative advantage of CAGE is more directly reflected in Table 2, while the $R/C$ curves mainly illustrate how compression affects conditional robustness, rather than relative improvement between methods.

---

> > ### Author Rebuttal · Reviewer_FEB3 · 2026-04-04
> >
> > I apprecaite the response and additional restuls presented by the authors. While most of my concerns have been addressed, my primary concern remains, i.e., the challenge itself and the effectiveness of CAGE. Either reducing the visual tokens from 576 to 192 or to 64 is a significant compression, and stronger compression like 16 (only 3\% of the visual token) could be impractical as it would also greatly undermine the benign performance of the model.

---

> > > ### Author Response · Authors · 2026-04-04
> > >
> > > Thanks for your further comments. We sincerely hope the following clarifications can address your concerns.
> > >
> > > > **Motivation**
> > >
> > > To address concerns about our motivation, we clarify the issue from the following three perspectives.
> > >
> > > 1. **The challenge persists under less extreme compression.**
> > > We agree that $K=16$ represents a relatively aggressive compression regime and may be too aggressive to serve as the primary motivating example. To address this concern, and following prior evaluations of token compression methods [1,2,3], which also report results at $K=32$, we additionally include results for $K=32$ as a less extreme yet still meaningful compression setting. As shown in Table 1, the mismatch gap remains evident even in this regime. For example, when $K_{\text{model}}=192$, using the misaligned setting $K_{\text{attack}}=32$ yields a robust accuracy of 58.3, whereas the aligned setting $K_{\text{attack}}=192$ reduces it to 50.7. This suggests that the challenge is not limited to only the most extreme compression levels.
> > >
> > > 2. **Aggressive compression settings remain meaningful evaluation regimes.**
> > > Although stronger compression can degrade benign performance, we believe such settings remain meaningful to evaluate because the model remains functional and these regimes are actively considered in recent token compression research. Even at $K_{\text{model}}=32$, VisionZip preserves **91.8%** of the original clean accuracy, indicating that the model does not collapse under this level of compression. More broadly, recent token compression methods explicitly explore increasingly smaller visual token budgets while maintaining acceptable utility [1,2,3,4]. Therefore, evaluating robustness in these regimes is still meaningful, rather than being limited to unrealistic failure cases.
> > >
> > > 3. **Aggressive compression can also be practically relevant.**
> > > Aggressive compression settings can still be relevant in latency-critical scenarios where some loss in accuracy is acceptable. For example, in real-time perception systems (e.g., robotics or autonomous driving), models operate under strict end-to-end latency constraints and process high-frequency visual streams. In these scenarios, system performance may depend less on preserving every fine-grained detail in an individual frame and more on maintaining fast and stable decisions over time, especially when temporal redundancy across frames can compensate for reduced single-frame accuracy. From this perspective, aggressively reducing computation, often the dominant bottleneck, is not merely a synthetic stress setting, but can reflect a realistic efficiency-driven operating regime.
> > >
> > > **Table 1.** Robust accuracy under different $K_{\text{model}}$ and $K_{\text{attack}}$.
> > >
> > > | $K_{\text{model}} \backslash K_{\text{attack}}$ | 576 (Full) | 192 | 64 | 32 | 16 |
> > > |---|---:|---:|---:|---:|---:|
> > > | 192 | 55.7 | **50.7** | 53.2 | 58.3 | 60.4 |
> > > | 64  | 53.5 | 48.8 | **48.7** | 54.3 | 56.0 |
> > > | 32  | 52.3 | 47.8 | 47.7 | **44.8** | 45.0 |
> > > | 16  | 49.7 | 45.3 | 44.7 | 44.6 | **44.4** |
> > >
> > > [1]Beyond Text-Visual Attention: Exploiting Visual Cues for Effective Token Pruning in VLMs, ICCV2025.
> > >
> > > [2]Beyond Attention or Similarity: Maximizing Conditional Diversity for Token Pruning in MLLMs, NeurIPS2025.
> > >
> > > [3]LearnPruner: Rethinking Attention-based Token Pruning in Vision Language Models, ICLR2026.
> > >
> > > [4]Delta-LLaVA: Base-then-Specialize Alignment for Token-Efficient Vision-Language Models, WACV2026.
> > >
> > > > Effectiveness
> > > 1.  While the absolute gain may vary by setting, we respectfully believe it is not minor: the reduction is consistent, often substantial in absolute robust accuracy, and reproducible across datasets, token budgets, and compression methods. For example, at $K=64$, the average robust accuracy drops from 31.1 to 18.9 on TextVQA ($-12.2$) and from 54.0 to 45.0 on VQA-v2 ($-9.0$). Importantly, this trend is observed across different compression methods (e.g., VisionZip, VisPruner, and FlowCut), rather than being limited to isolated cases.
> > > 2. Our key contribution is **not merely that CAGE produces a somewhat stronger attack, but that it shows conventional attacks can systematically overestimate robustness when token compression is applied**. Specifically, existing attacks optimize perturbations in the full-token space, whereas inference is actually performed after token pruning or merging. As a result, robustness can appear artificially high if the compression stage is ignored. In contrast, compression-aligned attacks expose substantially lower robust accuracy. This phenomenon is consistent across multiple token budgets, different compression mechanisms, and different LVLMs (e.g., LLaVA and Qwen), showing that the issue is systematic rather than anecdotal.
> > >
> > > Thank you again for your time and feedback. We sincerely hope these clarifications help address your remaining concerns.

---

### Decision · Program_Chairs · 2026-04-30

**Decision:**

Accept (regular)

**Comment:**

This paper investigates the adversarial robustness of Large Vision-Language Models (LVLMs) under visual token compression. The authors identify an "optimization-inference mismatch" in standard attacks and propose the **Compression-AliGnEd (CAGE) attack** as a more accurate evaluation tool. While initial reviews questioned the paper's scope and baseline comparisons , the authors provided a comprehensive rebuttal including additional models like InternVL3-8B, tasks such as image captioning and DriveQA, and multiple new baselines. This extensive experimentation successfully addressed concerns regarding depth and generalizability. The work is recognized as technically sound and highly relevant to the security-efficiency trade-offs in modern LVLMs. Consequently, the reviewers reached a consensus for acceptance.